# The spatiotemporal patterns of major human admixture events during the European Holocene

**Manjusha Chintalapati[1]\*, Nick Patterson[2,3]\*, Priya Moorjani[1,4]\***

[1]Department of Molecular and Cell Biology, University of California, Berkeley, Berkeley, United States; [2]Broad Institute of Harvard and MIT, Cambridge, United States; [3]Human Evolutionary Biology, Harvard University, Boston, United States; [4]Center for Computational Biology, University of California, Bekerley, Berkeley, United States

**Abstract** Recent studies have shown that admixture has been pervasive throughout human history. While several methods exist for dating admixture in contemporary populations, they are not suitable for sparse, low coverage ancient genomic data. Thus, we developed *DATES (Distribution of Ancestry Tracts of Evolutionary Signals)* that leverages ancestry covariance patterns across the genome of a single individual to infer the timing of admixture. *DATES* provides reliable estimates under various demographic scenarios and outperforms available methods for ancient DNA applications. Using *DATES* on ~1100 ancient genomes from sixteen regions in Europe and west Asia, we reconstruct the chronology of the formation of the ancestral populations and the fine-scale details of the spread of Neolithic farming and Steppe pastoralist-related ancestry across Europe. By studying the genetic formation of Anatolian farmers, we infer that gene flow related to Iranian Neolithic farmers occurred before 9600 BCE, predating the advent of agriculture in Anatolia. Contrary to the archaeological evidence, we estimate that early Steppe pastoralist groups (Yamnaya and Afanasievo) were genetically formed more than a millennium before the start of Steppe pastoralism. Our analyses provide new insights on the origins and spread of farming and Indo-European languages, highlighting the power of genomic dating methods to elucidate the legacy of human migrations.

**\*For correspondence:**
m_chintalapati@berkeley.edu (MC);
nickp@broadinstitute.org (NP);
moorjani@berkeley.edu (PM)

**Competing interest:** The authors declare that no competing interests exist.

## Editor's evaluation

This manuscript presents DATES, a robust method to infer the timing of admixture events using genetic data from present-day or ancient (with paleogenomics data) individuals. In the study, DATES is applied to >1000 ancient human genomes to characterize major admixture events during the European Holocene. This work will be of interest to scholars in the fields of population genetics, paleogenomics, archeology, biological anthropology, and history.

## Introduction

Recent studies have shown that population mixture (or 'admixture') is pervasive throughout human history, including mixture between the ancestors of modern humans and archaic hominins (i.e., Neanderthals and Denisovans), as well as in the history of many contemporary human groups such as African Americans, South Asians, and Europeans (*Pickrell and Reich, 2014*). Understanding the timing and signatures of admixture offers insights into the historical context in which the mixture occurred and enables the characterization of the evolutionary and functional impact of the gene flow. Many admixed groups are formed due to population movements involving ancient migrations that

predate historical records. The recent availability of genomic data for a large number of present-day and ancient genomes provides an unprecedented opportunity to reconstruct population events using genetic data, providing evidence complementary to linguistics and archaeology.

To characterize patterns of admixture, genetic methods use the insight that the genome of an admixed individual is a mosaic of chromosomal segments inherited from distinct ancestral populations (*Chakraborty and Weiss, 1988*). Due to recombination, these ancestral segments get shuffled in each generation and become smaller and smaller over time. The length of the segments is inversely proportional to the time elapsed since the mixture (*Chakraborty and Weiss, 1988*; *Moorjani et al., 2011*). Several genetic approaches – ROLLOFF (*Moorjani et al., 2011*; *Patterson et al., 2012*), ALDER (*Loh et al., 2013*), Globetrotter (*Hellenthal et al., 2014*), and Tracts (*Gravel, 2012*) – have been developed that use this insight by characterizing patterns of admixture linkage disequilibrium (LD) or haplotype lengths across the genome to infer the timing of mixture. Haplotype-based methods perform chromosome painting or local ancestry inference at each locus in the genome and characterize the distribution of ancestry tract lengths to estimate the time of mixture (*Gravel, 2012*; *Hellenthal et al., 2014*). This requires accurate phasing and inference of local ancestry, which is often difficult when the admixture events are old (as ancestry blocks become smaller over time) or when reference data from ancestral populations is unavailable. Admixture LD-based methods, on the other hand, measure the extent of the allelic correlation across markers to infer the time of admixture (*Loh et al., 2013*; *Moorjani et al., 2011*). They do not require phased data from the target or reference populations and work reliably for dating older admixture events (>100 generations). However, they tend to be less efficient in characterizing admixture events between closely related ancestral groups.

While highly accurate for dating admixture events using data from present-day samples, current methods do not work reliably for dating admixture events using ancient genomes. Ancient DNA samples often have high rates of DNA degradation, contamination (from human and other sources), and low sequencing depth, leading to a large proportion of missing variants and uneven coverage across the genome (*Orlando et al., 2021*). Additionally, most studies generate pseudo-haploid genotype calls – consisting of a haploid genotype determined by randomly selecting one allele at the variant site – that can lead to some issues in the inference. In such sparse datasets, estimating admixture LD can be noisy and biased (see Simulations below). Moreover, haplotype-based methods require phased data from both admixed and reference populations which remains challenging for ancient DNA specimens (*Gravel, 2012*; *Hellenthal et al., 2014*).

An extension of admixture LD-based methods, recently introduced by *Moorjani et al., 2016*, leverages ancestry covariance patterns that can be measured in a single sample using low coverage data. This approach measures the allelic correlation across neighboring sites, but instead of measuring admixture LD across multiple samples, it integrates data across markers within a single diploid genome. Using a set of ascertained markers that are informative for Neanderthal ancestry (where sub-Saharan Africans are fixed for the ancestral alleles and Neanderthals have a derived allele), *Moorjani et al., 2016*, inferred the timing of Neanderthal gene flow in Upper Paleolithic Eurasian samples and showed the approach works accurately in ancient DNA samples (*Moorjani et al., 2016*). However, this approach is inapplicable for dating admixture events within modern human populations, as there are very few fixed differences across populations (*Auton et al., 2015*).

Motivated by the single sample statistic in *Moorjani et al., 2016*, we developed *DATES (Distribution of Ancestry Tracts of Evolutionary Signals)* that measures the ancestry covariance across the genome in a single admixed individual, weighted by the allele frequency difference between two ancestral populations. This method was first introduced in *Narasimhan et al., 2019*, where it was used to infer the date of gene flow between groups related to Ancient Ancestral South Indians, Iranian farmers, and Steppe pastoralists in ancient South and Central Asian populations (*Narasimhan et al., 2019*). In this study, we evaluate the performance of *DATES* by carrying out extensive simulations for a range of demographic scenarios and comparing the approach to other published genomic dating methods. We then apply *DATES* to infer the chronology of the genetic formation of the ancestral populations of Europeans and the spatiotemporal patterns of admixture during the European Holocene using data from ~1100 ancient DNA specimens spanning ~8000–350 BCE.

## Results

### Overview of *DATES*: model and simulations

*DATES* estimates the time of admixture by measuring the weighted ancestry covariance across the genome using data from a single diploid genome and two reference populations (representing the ancestral source populations). *DATES* works like haplotype-based methods as it is applicable to a single genome and not like admixture LD-based methods, which by definition require multiple genomes to be co-analyzed; but unlike haplotype-based methods, it is more flexible as it does not require local ancestry inference. There are three main steps in *DATES*: we start by first learning the genome-wide ancestry proportions by performing a simple regression analysis to model the observed genotypes in an admixed individual as a linear mix of allele frequencies from two reference populations. For each marker, we then compute the likelihood of the observed genotype in the admixed individual using the estimated ancestry proportions and allele frequencies in each reference population (this is similar in spirit to local ancestry inference). This information is, in turn, used to compute the joint likelihood of shared ancestry at two neighboring markers, accounting for the probability of recombination between the two markers. Finally, we compute the covariance across pairs of markers located at a particular genetic distance, weighted by the allele frequency differences in the reference populations (Materials and methods).

Following *Moorjani et al., 2016*, we bin the markers that occur at a similar genetic distance across the genome, rather than estimating admixture LD for each pair of markers, and compute the covariance across increasing genetic distance between markers. The estimated covariance is expected to decay exponentially with genetic distance, and the rate of decay is informative of the time of the mixture (*Chakraborty and Weiss, 1988*; *Moorjani et al., 2011*). Assuming the gene flow occurred instantaneously, we can then infer the average date of gene flow by fitting an exponential distribution to the decay pattern (Materials and methods). In cases where data for multiple individuals is available, we compute the likelihood by summing over all individuals. To make *DATES* computationally tractable, we implement the fast Fourier transform (FFT) for calculating ancestry covariance as described in ALDER (*Loh et al., 2013*). This provides a speedup from $O\left(n^2\right)$ to $O\left(n \log n\right)$ that reduces the typical runtimes from hours to seconds with minimal loss in accuracy (*Appendix 1—figure 2*).

To assess the reliability of *DATES*, we performed simulations where we constructed 10 admixed diploid genomes by randomly sampling haplotypes from two source populations (Materials and methods). Briefly, we simulated individual genomes with 20% European and 80% African ancestry by using phased haplotypes of northern Europeans (Utah European Americans, CEU) and west Africans (Yoruba from Nigeria, YRI) from the 1000 Genomes Project, respectively (*Auton et al., 2015*). As reference populations in *DATES*, we used closely related surrogate populations of French and Yoruba respectively, from the Human Genome Diversity Panel (HGDP) (*Li et al., 2008*). We first investigated the accuracy of *DATES* by varying the time of admixture between 10 and 300 generations. For comparison, we also applied ALDER (*Loh et al., 2013*) to these simulations. Both methods reliably recovered the time of admixture up to 200 generations or ~5600 years ago, assuming a generation time of 28 years (*Moorjani et al., 2016*), though *DATES* was more precise than ALDER for older admixture events (>100 generations) (*Figure 1—figure supplement 1*, *Appendix 1—table 4*). Further, *DATES* shows accurate results even for single target samples (*Figure 1A*, *Figure 1—figure supplement 2A*) and even when few reference individuals are available for dating (*Figure 1A*, *Figure 1—figure supplement 2B*). However, the use of large numbers of reference samples, if available, can improve the inference. In *DATES*, allele frequencies of the reference populations are used for computing the likelihood as well as the weighted pairwise ancestry covariance across the genome (Materials and methods). With large samples, allele frequencies of the reference populations are more reliably computed, which in turn, can improve the precision of inferred dates (*Figure 1A*, *Figure 1—figure supplement 2B*).

Next, we tested *DATES* for features such as varying admixture proportions and use of surrogate populations as reference groups. By varying of European ancestry proportion between ~1% and 50% (the rest derived from west Africans), we observed *DATES* accurately estimated the timing in all cases (*Figure 1—figure supplement 3A*). However, the inferred admixture proportion was overestimated for lower admixture proportions (<10%) (*Figure 1—figure supplement 3B*). Thus, we caution against using *DATES* for estimating ancestry proportions. *DATES* works reliably for dating admixtures between related groups such as Europeans and Mexicans ($F_{ST}$~ 0.03), though it was unable to distinguish mixtures of Southern and Northern Europeans ($F_{ST}$< 0.005) (*Figure 1—figure supplement 5*).

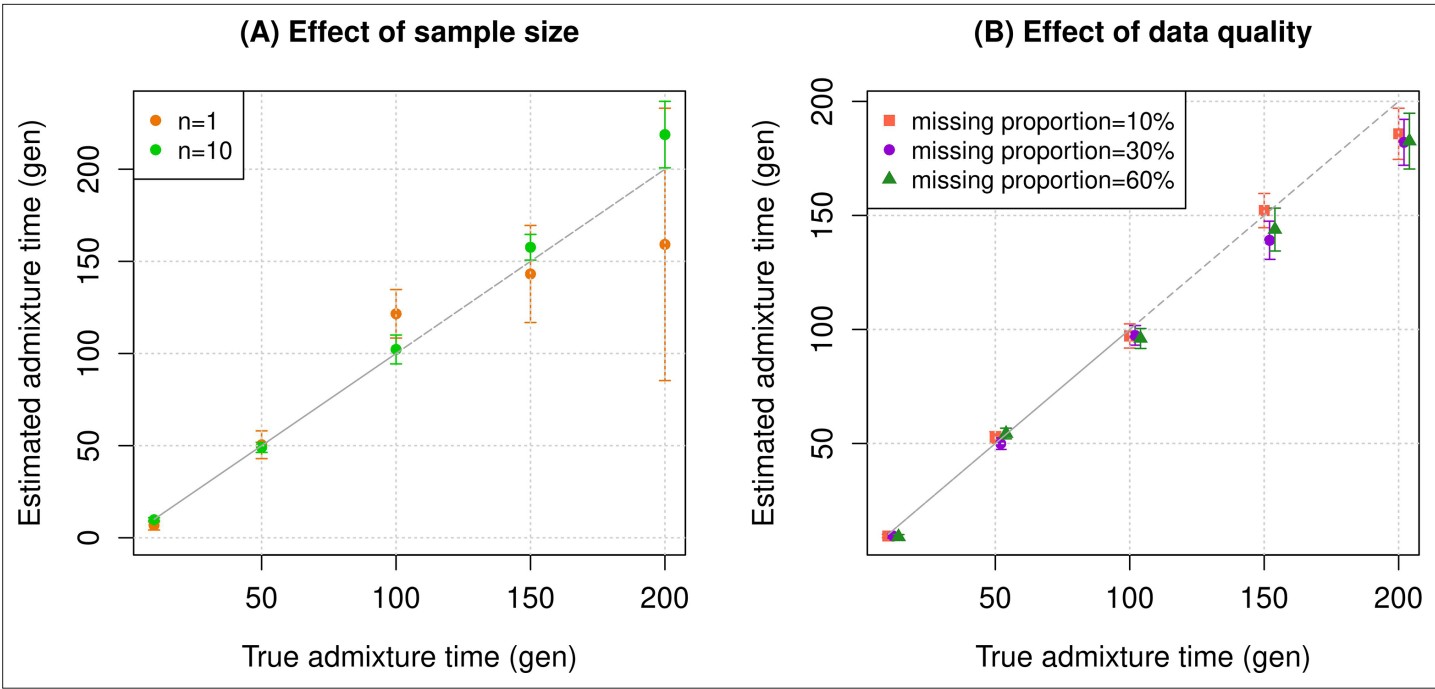

**Figure 1.** Simulation results. We constructed *n* admixed individuals with 20% European (CEU) and 80% African (YRI) ancestry using ~380,000 genome-wide SNPs for admixture dates ranging between 10 and 200 generations. To minimize any issues with overfitting, we used French and Yoruba from the Human Genome Diversity Panel as reference populations in *DATES (Distribution of Ancestry Tracts of Evolutionary Signals)*. We show the true time of admixture (X-axis, in generations) and the estimated time of admixture (±1 SE) (Y-axis, in generations). Standard errors were calculated using a weighted block jackknife approach by removing one chromosome in each run (Materials and methods). (**A**) Effect of sample size: We varied the sample size (*n*) of target group between 1 and 10 individuals. (**B**) Effect of data quality: To mimic the features of ancient genomes, we generated *n*=10 target individuals with pseudo-haploid genotypes and missing genotype rate as 10% (orange), 30% (purple), and 60% (green). See *Figure 1—figure supplements 1–9* for additional simulations to test the performance of DATES. R code to replicate this figure is available at: https://github.com/manjushachintalapati/DATES_EuropeanHolocene/blob/main/1.R.

The online version of this article includes the following figure supplement(s) for figure 1:

**Figure supplement 1.** Varying time of admixture up to 300 generations.

**Figure supplement 2.** Impact of sample size of the target (admixed) and reference populations.

**Figure supplement 3.** Impact of admixture proportion.

**Figure supplement 4.** Impact of divergence between the ancestral population and reference populations used in *DATES (Distribution of Ancestry Tracts of Evolutionary Signals)*.

**Figure supplement 5.** Impact of divergence between the two source populations.

**Figure supplement 6.** Impact of using the admixed individuals themselves as one of the reference groups in *DATES (Distribution of Ancestry Tracts of Evolutionary Signals)*.

**Figure supplement 7.** Impact of sample size and data quality of target samples.

**Figure supplement 8.** Impact of data quality of target and reference populations as a function of divergence between true and reference populations used in *DATES (Distribution of Ancestry Tracts of Evolutionary Signals)*.

**Figure supplement 9.** Impact of small sample size and data quality of target and reference populations as a function of divergence between true and reference populations used in *DATES (Distribution of Ancestry Tracts of Evolutionary Signals)*.

We found *DATES* is robust to the use of highly divergent surrogates as reference populations. For example, the use of Khomani San as the reference population instead of the true ancestral population of Yoruba ($F_{ST}$ ~ 0.1) provides unbiased dates of admixture (*Figure 1—figure supplement 4*). In this regard, for ancient DNA where sometimes only sparse data is available, one can also use present-day samples as reference populations to increase the quality and sample size of the ancestral groups. In principle, as long as the allele frequencies in the reference samples are correlated to the ancestral allele frequencies, the inference of admixture dates should remain unbiased (Materials and methods). In practice, however, recent demographic events (e.g., strong founder events or admixture

from additional sources, etc.) in the history of the present-day samples could lead to significant deviation from the ancestral allele frequencies. Thus, the reference populations should be carefully chosen.

Another idea is to use the admixed populations themselves as one of the reference populations as demonstrated by the single reference setup in ALDER (*Loh et al., 2013*). Admixed individuals have intermediate allele frequencies to the ancestral populations and thus weighted LD or ancestry covariance can be computed with only one reference population (albeit, with reduced power). *Loh et al., 2013*, showed that the use of admixed populations as one of the references does not bias the rate of decay of the weighted LD (i.e., time of admixture), though the amplitude of the decay curve (not used in *DATES*) can be biased under some scenarios. To verify *DATES* provides reliable results under this setup, we applied *DATES* with a single reference population and used the admixed population as the other reference. Like ALDER, our inferred dates of admixture were accurate and comparable to using two reference populations. (*Figure 1—figure supplement 6*).

An important feature of *DATES* is that it does not require phased data and is applicable to datasets with small sample sizes, making it in principle useful for ancient DNA applications. To test the reliability of *DATES* for ancient genomes, we simulated data mimicking the relevant features of ancient genomes, namely small sample sizes ($n$=1–20), large proportions of missing genotypes (between 10% and 60%), and pseudo-haploid genotype calls (instead of diploid genotype calls) in reference and/or target samples. *DATES* showed reliable results under various setups, even when only a single admixed individual was available (*Figure 1B*, *Figure 1—figure supplements 1–9*). In contrast, admixture LD-based methods require more than one sample and do not work reliably with missing data. For example, ALDER estimates were very unstable for simulations with >40% missing data. For older dates (>100 generations), there was a slight bias even with >10% missing genotypes (*Appendix 1—figure 5*). This is expected as LD calculations leverage shared patterns across samples, thus variable missingness of genotypes across individuals leads to substantial loss of data leading to unstable and noisy inference. We also generated data for combinations of features including small sample sizes, pseudo-haploid genotypes with large proportions of missing genotypes in both target and reference samples, and use of highly divergent reference samples. We found *DATES* yielded reliable results with large amounts (~40–60%) of missing data, either in the target or references, even with highly divergent reference populations (*Figure 1—figure supplement 8*). This was also true when a single target sample was available, though as expected, the inference becomes noisier for older dates and large fractions of missing data (*Figure 1—figure supplement 9*). The robust performance of *DATES* in sparse datasets highlights a major advantage for ancient DNA applications.

*DATES* assumes a model of instantaneous gene flow with a single pulse of mixture between two source populations. However, many human populations have a history of multiple pulses of gene flow. To test the performance of *DATES* for multi-way admixture events, we generated admixed individuals with ancestry from three sources (East Asians, Africans, and Europeans) where the gene flow occurred at two distinct time points (*Appendix 2—figure 1*). By applying *DATES* with pairs of reference populations, we observed that *DATES* recovered both admixture times for target populations that had equal contributions from all three ancestral groups (*Appendix 2—figure 2*). In the case of unequal admixture proportions from three ancestral groups, *DATES* inferred the timing of the recent admixture event in most cases. In some cases, however, the inferred dates were intermediate to the two pulses when the ancestry proportion of the recent event was low (*Appendix 2—figure 3*). This confounding could be eliminated if the reference populations were set up to match the model of gene flow. For example, the inferred times of admixture were accurate if the two references used in *DATES* were: reference 1: the source population for the recent event and reference 2: pooled individuals from both ancestral populations that contributed to the first admixture event, or the intermediate admixed group formed after the first event (*Appendix 2—table 1*). This highlights how the choice of reference populations can help to tune the method to infer the timing of specific admixture events more reliably.

Finally, we explored the impact of more complex demographic events, including continuous admixture and founder events using coalescent simulations (Appendix 2). In the case of continuous admixture, *DATES* inferred an intermediate timing between the start and the end of the gene flow period, similar to other methods like ALDER and Globetrotter (*Hellenthal et al., 2014*; *Loh et al., 2013*; *Appendix 2—table 2*). In the case of populations with founder events, we inferred unbiased dates of admixture in most cases except when the founder event was extreme ($N_e$ ~ 10) or the population had maintained a low population size ($N_e$ < 100) until the present (i.e., no recovery bottleneck)

(*Appendix 2—figure 4*, *Appendix 2—table 3*). In humans, few populations have such extreme founder events, and thus, in most other cases, our inferred admixture dates should be robust to founder events (*Tournebize et al., 2020*). We note that while *DATES* is not a formal test of admixture, in simulations, we find that in the absence of gene flow, the method does not infer significant dates of admixture even if the target has a complex demographic history (*Appendix 2—figure 6*, *Appendix 2—figure 7*).

## Comparison to other methods

We assessed the reliability of *DATES* in real data by comparing our results with published methods: Globetrotter, ALDER, and ROLLOFF. These methods are designed for the analysis of present-day samples that typically have high-quality data with limited missing variants. In addition, Globetrotter uses phased data which is challenging for ancient DNA samples. Thus, instead of rerunning other methods, we took advantage of the published results for contemporary samples presented in *Hellenthal et al., 2014*. Following *Hellenthal et al., 2014*, we created a merged dataset including individuals from HGDP (*Li et al., 2008*, *Behar et al., 2010*, and *Henn et al., 2012*) (Materials and methods). We applied *DATES* and ALDER to 29 target groups using the reference populations reported in Table S12 in *Hellenthal et al., 2014*, excluding one group where the population label was unclear. Interestingly, the majority of these groups (25/29) failed ALDER's formal test of admixture; either because the results of the single reference and two reference analyses yielded inconsistent estimates or because the target had long-range shared LD with one of the reference populations (*Appendix 1—table 5*). Using *DATES*, we inferred significant dates of admixture in 20 groups, and 14 of those were consistent with estimates based on Globetrotter. In the case of the six populations that disagreed across the two methods, most of the populations appear to have a history of multiple pulses of gene flow either involving more than two populations (e.g., Brahui *Pagani et al., 2017*) or multiple instances of contact between the same two reference groups (e.g., Mandenka *Price et al., 2009*) or the model of admixture differed (e.g., recent ancient DNA studies suggest present-day Bulgarians have ancestry from western hunter-gatherers [HGs], Near Eastern farmers, and Steppe pastoralists from Eurasia [*Haak et al., 2015*] but were modeled as a mixture of Polish and Cypriots in Globetrotter). In case of complex admixture scenarios, the inconsistencies across the two methods are hard to interpret as Globetrotter and *DATES* could be capturing different events or the weighting of both events could differ. Finally, the estimated admixture timing based on *DATES*, ROLLOFF, and ALDER (assuming two-way admixture regardless of the formal test results) were found to be highly concordant (*Appendix 1—table 5*).

## Fine-scale patterns of population mixtures in ancient Europe

Recent ancient DNA studies have shown that present-day Europeans derive ancestry from three distinct sources: (a) HG-related ancestry that is closely related to Mesolithic HGs from Europe; (b) Anatolian farmer-related ancestry related to Neolithic farmers from the Near East and associated to the spread of farming to Europe; and (c) Steppe pastoralist-related ancestry that is related to the Yamnaya pastoralists from Russia and Ukraine (*Allentoft et al., 2015*; *Haak et al., 2015*; *Skoglund et al., 2012*). Many open questions remain about the timing and dynamics of these population interactions, in particular related to the formation of the ancestral groups (which were themselves admixed) and their expansion across Europe. To characterize the spatial and temporal patterns of mixtures in Europe in the past 10,000 years, we used 1096 ancient European samples from 152 groups from the publicly available Allen Ancient DNA Resource (AADR) spanning a time range of ~8000–350 BCE (Materials and methods, *Supplementary file 1A*). Using *DATES*, we characterized the timing of the various gene flow events, and below, we describe the key events in chronological order focusing on three main periods.

### Holocene to Mesolithic

Pre-Neolithic Europe was inhabited by HGs until the arrival of the first farmers from the Near East (*Haak et al., 2010*; *Hofmanová et al., 2016*). There was large diversity among HGs with four main groups – western hunter-gatherers (WHGs) that were related to the Villabruna cluster in central Europe, eastern hunter-gatherers (EHGs) from Russia and Ukraine related to the Upper Paleolithic group of Ancestral North Eurasians (ANEs), Caucasus hunter-gatherers (CHGs) from Georgia associated to the first farmers from Iran, and the GoyetQ2-cluster associated to the Magdalenian culture in Spain and

Portugal (*Fernandes et al., 2018*; *Fu et al., 2016*; *Jones et al., 2015*; *Rivollat et al., 2020*; *Skoglund et al., 2012*). Most Mesolithic HGs fall on two main clines of relatedness: one cline that extends from Scandinavia to central Europe showing variable WHG-EHG ancestry, and the other in southern Europe with WHG-GoyetQ2 ancestry (*Rivollat et al., 2020*). The latter is already present in the 17,000 BCE *El Mirón* individual from Spain, suggesting that the GoyetQ2-related gene flow occurred well before the Holocene. However, the WHG-EHG cline was formed more recently during the Mesolithic period, though the precise timing remains less well understood.

To characterize the formation of the WHG-EHG cline, we used genomic data from 16 ancient HG groups (*n*=101) with estimated ages of ~7500–3600 BCE. We first verified the ancestry of each HG group using *qpAdm* that compares the allele frequency correlations between the target and a set of source populations to formally test the model of admixture and then infer the ancestry proportions for the best-fitted model (*Haak et al., 2015*). For each target population, we chose the most parsimonious model, that is, fitting the data with the minimum number of source populations. Consistent with previous studies, our *qpAdm* analysis showed that most HGs from Scandinavia, the Baltic Sea region, and central Europe could be modeled as a two-way mixture of WHG- and EHG-related ancestry (*Supplementary file 2A*). To confirm that the target populations do not harbor Anatolian farmer-related ancestry (that could lead to some confounding in estimated admixture dates), we applied *D*-statistics of the form *D*(Mbuti, *target*, WHG, Anatolian farmers) where target = Mesolithic HGs. We observed that none of the target groups had a stronger affinity to Anatolian farmers than WHG (*Supplementary file 2B*). Together, these results suggest that the mixtures we date below reflect pre-Neolithic contacts between the HGs.

To infer the timing of the mixtures in the history of Mesolithic European HGs, we applied *DATES* to HGs from Scandinavia, the Baltic regions, and central Europe using WHG- and EHG-related groups as reference populations. *DATES* infers the time of admixture in generations before the sample lived. Accounting for the average sampling age of the specimens and the mean human generation time of 28 years (*Moorjani et al., 2016*), we inferred the admixture time in years before present or in BCE (Materials and methods). We report the average dates (or median, where specified) in BCE in the main text and provide additional details in *Figure 2* and *Supplementary file 1B* including the sample sizes, dates in generations, and BCE for each population. Among HGs, we inferred that the earliest admixture occurred in Scandinavian HGs from Norway and Sweden with a range of average dates of ~80–113 generations before the samples lived (*Figure 2—figure supplement 1*). This translates to admixture dates of ~10,200–8000 BCE, with the most recent dates inferred in Motala HGs from Sweden suggesting substantial substructure in HGs (*Figure 2*). In the Baltic region, we inferred the range of admixture dates of ~8700–6000 BCE in Latvia and Lithuania HGs, postdating the mixture in Scandinavia. In southeast Europe, the Iron Gates region of the Danube Basin shows widespread evidence of mixtures between HG groups and, in the case of some outliers, the mixture of HGs and Anatolian farmer-related ancestry as early as the Mesolithic period (*Feldman et al., 2019*). Further, these groups showed a strong affinity to the WHG-related ancestry in Anatolian populations, suggesting ancient interactions with Near Eastern populations (*Feldman et al., 2019*). We applied *qpAdm* to test the model of admixture in Iron Gates HG and found that the parsimonious model with WHG- and EHG-related ancestry provides a good fit to the data. Further, when we tested the model with Anatolian-related ancestry using Anatolian HG (AHG) as an additional source population, the AHG ancestry proportion was not significant (*Supplementary file 2A*). Applying *DATES* to Iron Gates HG with WHG and EHG as reference populations, we inferred this group was genetically formed in ~9200 BCE (95% confidence interval: 10,000–8400 BCE). Our samples of the Iron Gates HGs include a wide range of C14 dates between 8800 and 5700 BCE. We confirmed our dates were robust to the sampling age of the individuals as we obtained statistically consistent dates when all samples were combined as one group or when subsets of samples were grouped in bins of 500 years (*Figure 2—figure supplement 2*). The most recent dates of ~7500 BCE were inferred in eastern Europe in Ukraine HGs, highlighting how the WHG-EHG cline was formed over a period ~2000–3000 years (*Figure 2*, *Supplementary file 1B*).

## Early to middle Neolithic

Neolithic farming began in the Near East – the Levant, Anatolia, and Iran – and spread to Europe and other parts of the world (*Haak et al., 2010*; *Kılınç et al., 2016*; *Skoglund et al., 2012*). The first

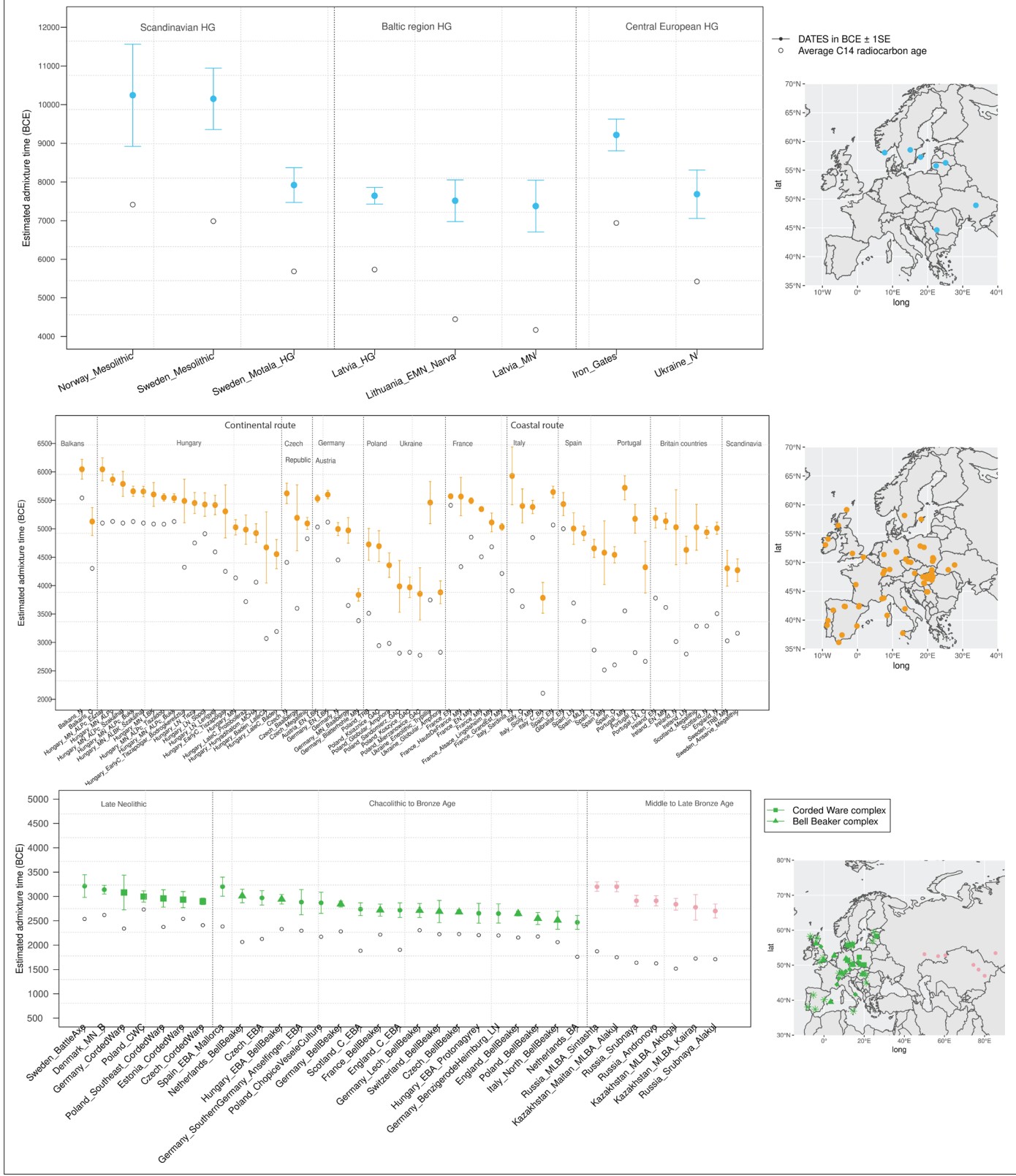

**Figure 2.** Timeline of admixture events in ancient Europe. We applied *DATES (Distribution of Ancestry Tracts of Evolutionary Signals)* to ancient samples from Europe. In the right panel, we show the sampling locations of the ancient specimens, and in the left panel, we show the admixture dates for each target group listed on the X-axis. The inferred dates in generations were converted to dates in BCE by assuming a mean generation time of 28 years (*Moorjani et al., 2011*) and accounting for the average sampling age (shown as gray dots) of all ancient individuals in the target group (Materials and

*Figure 2 continued on next page*

*Figure 2 continued*

methods). The top panel shows the formation of western hunter-gatherer (WHG)-eastern hunter-gatherer (EHG) cline (in blue) using Mesolithic hunter-gatherers (HGs) as the target and EHG and WHG as reference populations. The middle panel shows admixture dates of local HGs and Anatolian farmers (in orange) using Neolithic European groups as targets and Anatolian farmers-related groups and WHG-related groups as reference populations. The bottom panel shows the spread of Steppe pastoralist-related ancestry (in green) estimated using middle and late Neolithic, Chalcolithic, and Bronze Age samples from Europe as target populations and early Steppe pastoralist-related groups (Afanasievo and Yamnaya Samara) and a set of Anatolian farmers and WHG-related groups as reference populations. For the middle to late Bronze Age (MLBA) samples from Eurasia, we used the early Steppe pastoralist-related groups and the Neolithic European groups as reference populations. The cultural affiliation (Corded Ware Complex [CWC], Bell Beaker complex [BBC], or Steppe MLBA cultures) of the individuals is shown in the legend. See *Figure 2—figure supplements 1 and 2* we applied DATESfor decay curves for all samples and stratified datesfor Iron Gates HGs. R code to replicate this figure is available at: https://github.com/manjushachintalapati/DATES_EuropeanHolocene/blob/main/3.R.

The online version of this article includes the following figure supplement(s) for figure 2:

**Figure supplement 1.** *DATES (Distribution of Ancestry Tracts of Evolutionary Signals) ancestry covariance decay curves.*

**Figure supplement 2.** Timing of western hunter-gatherer (WHG) and eastern hunter-gatherer (EHG) admixture in Iron Gates hunter-gatherer (HG) samples.

farmers of Europe were related to Anatolian farmers, whose origin remains unclear. The early Neolithic Anatolian farmers (Aceramic Anatolian farmers) had majority ancestry from AHG with some gene flow from the first farmers from Iran (*Feldman et al., 2019*). AHG, in turn, had ancestry from Levant HG (Natufians) and some mysterious HG group related to the ancestors of WHG individuals from central Europe – a gene flow event that likely occurred in the late Pleistocene (*Feldman et al., 2019*). Using *qpAdm*, we confirmed that early Anatolian farmers could be modeled as a mixture of AHG and Iran Neolithic farmer-related groups (*Supplementary file 2C*). To learn about the timing of the genetic formation of early Anatolian farmers, we applied *DATES* using Iran Neolithic farmer-related individuals and other reference as groups with AHG ancestry. Since there are limited samples of AHG ancestry, we instead used pooled individuals of WHG-related and Levant Neolithic farmer-related individuals to represent the main ancestry components of AHG. We note that the application of *DATES* to three-way admixed groups such as early Anatolian farmers can lead to intermediate dates between the first and second pulse of gene flow unless the reference populations are chosen carefully (*Appendix 2—table 1*). Our setup with pooled reference populations should recover the timing of the most recent event (in this case, the gene flow from CHG or Iran Neolithic-related groups) reliably. We infer the Iran Neolithic farmer-related gene flow occurred ~10,900 BCE (12,200–9600 BCE) (*Figure 3*), predating the origin of farming in Anatolia (*Bramanti et al., 2009*). During the subsequent millennia, these early farmers further admixed with Levant Neolithic groups to form Anatolian Neolithic farmers who spread towards the west to Europe and in the east to mix with Iran Neolithic farmers, forming the Chalcolithic groups of Seh Gabi and Hajji Firuz (*Supplementary file 2C*). Using *DATES*, we inferred that these Chalcolithic groups were genetically formed in ~7600–5700 BCE (*Supplementary file 1B*).

In Europe, the Anatolian Neolithic farmers mixed with the local indigenous HGs contributing between ~40% and 98% of ancestry to the Neolithic Europeans. To elucidate the fine-scale patterns and regional dynamics of these mixtures, we applied *DATES* to time transect samples from 94 groups (*n*=657) sampled from 16 regions in Europe, ranging from ~6000 to 1900 BCE and encompassing individuals from the early Neolithic to Chalcolithic periods (*Supplementary file 1A*). Using *qpAdm*, we first confirmed that the Neolithic Europeans could be modeled as a mixture of European HG-related ancestry and Anatolian farmer-related ancestry and inferred their ancestry proportions (*Supplementary file 2D*). For most target populations (~80%), we found the model of gene flow between Anatolian farmer-related and WHG-related ancestry provided a good fit to the data (p-value > 0.05). In some populations, we found variation in the source of the HG-related ancestry and including either EHG- or GoyetQ2-related ancestry groups improved the fit of the model. In five groups, none of the models fit, despite excluding outlier individuals whose ancestry profile differed from the majority of the individuals in the group (*Supplementary file 2E*). To confirm that the target populations do not harbor Steppe pastoralist-related ancestry, we applied *D*-statistics of the form *D*(Mbuti, *target*, Anatolian farmers, Steppe pastoralists) where target = Neolithic European groups. We observed that four groups had a stronger affinity to Steppe pastoralists compared to Anatolian farmers, and hence we excluded these from further analysis (*Supplementary file 2F*). After filtering, we applied *DATES*

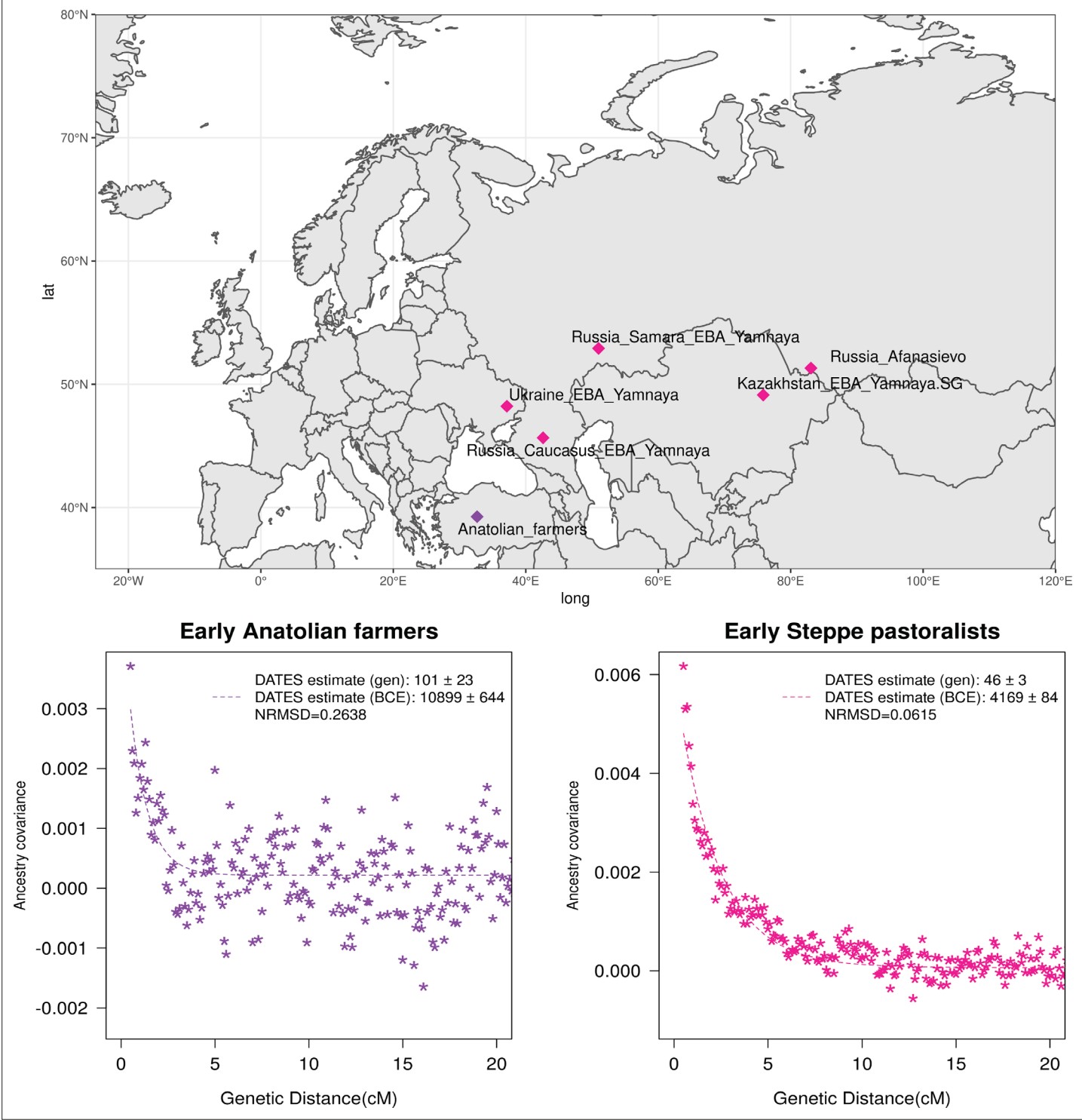

**Figure 3.** Genetic formation of early Anatolian farmers and early Bronze Age Steppe pastoralists. The top panel shows a map with sampling locations of the target groups analyzed for admixture dating. The bottom panels show the inferred times of admixture for each target using *DATES (Distribution of Ancestry Tracts of Evolutionary Signals)* by fitting an exponential function with an affine term $y = Ae^{-\lambda d} + c$, where $d$ is the genetic distance in Morgans and $\lambda = (t+1)$ is the number of generations since admixture ($t$) (Materials and methods). We start the fit at a genetic distance ($d$) >0.5 cM (centiMorgans) to minimize confounding with background LD and estimate a standard error by performing a weighted block jackknife removing one chromosome in each run. For each target, in the legend, we show the inferred average dates of admixture (±1 SE) in generations before the individual lived, in BCE accounting for the average age of all the individuals and the mean human generation time, and the normalized root-mean-square deviation (NRMSD) values to assess the fit of the exponential curve (Materials and methods). The bottom left shows the ancestry covariance decay curve for early Anatolian

*Figure 3 continued on next page*

*Figure 3 continued*

farmers inferred using one reference group as a set of pooled individuals of western hunter-gatherer (WHG)-related and Levant Neolithic farmers-related individuals as a proxy of Anatolian hunter-gatherer (AHG) ancestry and the second reference group containing Iran Neolithic farmer-related individuals. The bottom right shows the ancestry covariance decay curve for early Steppe pastoralists groups, including all Yamnaya and Afanasievo individuals as the target group and eastern hunter-gatherer (EHG)-related and Iran Neolithic farmer-related groups as reference populations. R code to replicate this figure is available at: https://github.com/manjushachintalapati/DATES_EuropeanHolocene/blob/main/2.R.

to 86 European Neolithic groups using WHG-related individuals and Anatolian farmers as reference populations.

Earlier analysis has suggested that farming spread along two main routes in Europe, from southeast to central Europe ('continental route') and along the Mediterranean coastline to Iberia ('coastal route') (*Gronenborn, 2014*; *Guilaine, 2003*; *Rivollat et al., 2020*). Consistent with this, we inferred one of the earliest timings of gene flow was in the Balkans around 6400 BCE. Using the most comprehensive time-transect in Hungary with 19 groups (*n*=63) spanning from middle Neolithic to late Chalcolithic, we inferred the admixture dates ranged from ~6100 to 4500 BCE. Under a model of a single shared gene flow event in the common ancestors of all individuals, we would expect to obtain similar dates of admixture (before present) after accounting for the age of the ancient specimens. Similar to *Lipson et al., 2017*, we observed that the estimated dates in middle Neolithic individuals were substantially older than those inferred in late Neolithic or Chalcolithic individuals (*Bollongino et al., 2013*). This would be expected if the underlying model of gene flow involved multiple pulses of gene flow, such that the timing in the middle Neolithic samples reflects the initial two-way mixture and the timing in the Chalcolithic samples captures both recent and older events. Interestingly, *Lipson et al., 2017*, and other recent studies have documented increasing HG ancestry from ~3% to 15% from the Neolithic to Chalcolithic period (*Haak et al., 2015*; *Lipson et al., 2017*; *Rivollat et al., 2020*), suggesting that there was additional HG gene flow after the initial mixture. This highlights that the interactions between local HGs and incoming Anatolian farmers were complex with multiple gene flow events or continuous admixture between these two groups, which explains the increasing HG ancestry and more recent dates in Chalcolithic individuals (*Supplementary file 2D*).

Mirroring the pattern in Hungary, we documented the resurgence of HG ancestry in the Czech Republic, France, Germany, and southern Europe. In central Europe, we inferred that the Anatolian farmer-related gene flow ranged between ~5600 and 5000 BCE across Germany and Czech Republic, with some exceptions. For instance, in the Blätterhöhle site from Germany, the inferred dates were more recent (~4000 BCE), consistent with the occupation of both HGs and farmers in this region until the late Neolithic (*Lipson et al., 2017*). In eastern Europe, using samples related to the Funnel Beaker culture (TRB; from German *Trichterbecher*) from Poland, we dated the Anatolian farmer-related gene flow occurred on average ~4700 BCE (5300–4200 BCE). Following the TRB decline, the Baden culture and the Globular Amphora culture appeared in many areas of Poland and Ukraine (*Fernandes et al., 2018*). These cultures had close contact with the Corded Ware complex (CWC) and Steppe pastoralists' societies, though we found a parsimonious model without Steppe pastoralist-related ancestry provides a good fit to the GAC individuals (*Supplementary file 2D*). Applying *DATES*, we inferred the Anatolian farmer and HG-related mixture in GAC ranged between ~4700 and 3900 BCE, predating the spread of Steppe pastoralists to eastern Europe (*Allentoft et al., 2015*; *Haak et al., 2015*).

Along the Mediterranean route, we characterized Anatolian farmer-related gene flow in Italy, Iberia, France, and the British Isles. Using samples from five groups in Italy, we inferred the earliest dates of gene flow of ~6100 BCE, and within the millennium, the Anatolian farmer-related ancestry spread from Sardinia to Sicily (*Figure 2*). In Iberia, the Anatolian farmer-related mixture ranged from ~5700 to 4300 BCE and showed evidence for an increase in HG ancestry from ~9% to 20% after the initial gene flow. In France, previous studies have shown that Anatolian farmer-related ancestry came from both routes, along the continental route in the north and along the costal route in the south (*Rivollat et al., 2020*). This is reflected in the source of the HG ancestry, which is predominantly EHG and WHG-related in the north and includes WHG and Goyet-Q2 ancestry in the south (*Rivollat et al., 2020*). Consistently, we also observed that the admixture dates in France were structured along these routes, with the median estimate of ~5100 BCE in the east and much older ~5500 BCE in the south (*Supplementary file 1B*). In Scandinavia, we inferred markedly more recent dates of admixture of ~4300 BCE

using samples from Sweden associated with the TRB culture and Ansarve Megalithic tombs, consistent with a late introduction of farming to Scandinavia (**Mittnik et al., 2018**).

Finally, we inferred recent dates of admixture in Neolithic samples from the British Isles (England, Scotland, and Ireland) with the median timing of ~5000 BCE across the three regions. Interestingly, unlike in western and southern Europe, we obtained overlapping dates across eight groups including early to late Neolithic samples from British Isles. This is consistent with previous studies that suggest there was no resurgence in HG ancestry during the Neolithic in Britain (**Brace et al., 2019**). Thus our dates can be interpreted as the time of the main mixture of HGs and Anatolian farmers in this region, implying that the farmer-related ancestry reached Britain a millennium after its arrival in continental Europe. By 4300 BCE, we find that Anatolian farmer-related ancestry is present in nearly all regions in Europe.

## Late Neolithic to Bronze Age

The beginning of the Bronze Age (BA) was a period of major cultural and demographic change in Eurasia, accompanied by the spread of Yamnaya Steppe pastoralist-related ancestry from Pontic-Caspian steppes across Europe and South Asia (**Haak et al., 2015**). The archaeological record documents that the early Steppe pastoralists cultures of Yamnaya and Afanasievo, with characteristic burial styles and pottery, appeared ~3300–2600 BCE (**Morgunova and Khokhlova, 2016**). These groups were formed as a mixture of EHG-related individuals and CHG-related groups associated with the first farmers from Iran (**Jones et al., 2015**; **Narasimhan et al., 2019**; **Wang et al., 2019**). Using *qpAdm*, we first tested how well this model fits the data from eight early Steppe pastoralist groups, including seven groups associated with Yamnaya culture and one group related to the Afanasievo culture (Materials and methods). For all but two Yamnaya groups (from Hungary Baden and Russia Kalmykia), we found this model provides a good fit to the data (**Supplementary file 2G**). We note that the samples from Kalmykia in our dataset were shotgun sequenced, and in the *qpAdm* analysis, we are mixing shotgun and capture data that could potentially lead to technical issues. To understand the timing of the formation of the early Steppe pastoralist-related groups, we applied *DATES* using pooled EHG-related and pooled Iranian Neolithic farmer-related individuals. Focusing on the groups with the largest sample sizes, Yamnaya Samara ($n$=10) and Afanasievo ($n$=19), we inferred the admixture occurred between 40 and 45 generations before the individuals lived, translating to an admixture timing of ~4100 BCE (**Supplementary file 1B**). We obtained qualitatively similar dates across four Yamnaya and one Afanasievo groups, consistent with the findings that these groups descend from a recent common ancestor (we note for the Ozera samples from Ukraine, the dates were not significant). This is also further supported by the insight that the genetic differentiation across early Steppe pastoralist groups is very low ($F_{ST} \sim$ 0.000–0.006) (**Supplementary file 2H**). Thus, we combined all early Steppe pastoralist individuals in one group to obtain a more precise estimate for the genetic formation of proto-Yamnaya of ~4400–4000 BCE (**Figure 3**). These dates are noteworthy as they predate the archaeological evidence by more than a millennium (**Anthony, 2007**) and have important implications for understanding the origin of proto-Pontic Caspian cultures and their spread to Europe and South Asia.

Over the following millennium, the Yamnaya-derived ancestry spread across Europe through CWC and Bell Beaker complex (BBC) cultures. Present-day Europeans derive between ~10% and 60% Steppe pastoralist-related ancestry, which was not seen in Neolithic samples. To obtain a precise chronology of the spread of Steppe pastoralist-related ancestry across Europe, we analyzed 109 late Neolithic, Chalcolithic, and BA samples dated between 3000 and 750 CE from 18 regions, including samples associated with the CWC and BBC cultures. We first confirmed that most target samples had Steppe pastoralist-related ancestry, in addition to European HG-related and Anatolian farmer-related ancestry using *qpAdm*. We excluded 20 groups that could not be parsimoniously modeled as a three-way mixture even after removing individual outliers. After filtering, we retained 79 groups for dating Steppe pastoralist-related gene flow across Europe (**Supplementary file 2I and J**). As BA Europeans have ancestry from three distinct groups, we applied *DATES* using the following two reference populations, one group including early Steppe pastoralists (Yamnaya and Afanasievo) and the other group that is a the proxy for the ancestral Neolithic Europe population using pooled samples of WHG-related and Anatolian farmer-related individuals.

To learn about the spread of CWC culture across Europe, we used seven late Neolithic and Bronze age groups, including five associated with CWC artifacts. Using *DATES*, we inferred that the oldest date of Steppe pastoralists gene flow in Europe was ~3200 BCE in Scandinavia in samples associated with Battle Axe Culture in Sweden and Single Grave Culture in Denmark that were both contemporary to CWC. The samples from Scandinavia showed large heterogeneity in ancestry, including some individuals with majority Steppe pastoralist-related ancestry (and negligible amounts of Anatolian farmer-related ancestry), consistent with patterns expected from recent gene flow (*Malmström et al., 2019*). Strikingly, we inferred the timing of admixture in central Europe (Germany and the Czech Republic) and eastern Europe (Estonia and Poland) to be remarkably similar. These dates fall within a narrow range of ~3000–2900 BCE across diverse regions, suggesting that the mixed population associated with the Corded Ware culture formed over a short time and spread across Europe rapidly with very little further mixture (*Supplementary file 1B*).

Following the Corded Ware culture, from around 2800 to 2300 BCE, Bell Beaker pottery became widespread across Europe (*Fokkens and Nicolis, 2012*). Using 19 Chalcolithic and BA samples, including 10 associated with Beaker-complex artifacts, we inferred the dynamics of the spread of the Beaker complex across Europe. We inferred the oldest date of Steppe pastoralist-related admixture was ~3200 BCE (3600–2800 BCE) in early Bronze Age (EBA) Mallorca samples from Iberia. We note the EBA Mallorca sample is not directly associated with Beaker culture, but *qpAdm* modeling suggests that this individual is clade with the small subset of Iberian Beaker-complex-associated individuals who carried Steppe pastoralist-related ancestry (*Fernandes et al., 2020*). Most individuals from Iberia, however, had negligible Steppe pastoralist-related ancestry suggesting the Beaker culture was not accompanied by major gene flow in Iberia despite the earliest dates (*Supplementary file 2I*). In central and western Europe, where Steppe pastoralist gene flow was more pervasive, we inferred the median date of the mixture was ~2700 BCE with the oldest dates in the Netherlands, followed by Germany and France (*Figure 2*). There was, however, large heterogeneity in the dates across Europe and even within the same region. For example, comparing two BA groups from the Netherlands suggests a wide range of dates ~3000 BCE and 2500 BCE, and four groups from Germany indicate a range of ~2900–2700 BCE. From central Europe, the Steppe pastoralist-related ancestry spread quickly to the British Isles, where people with Steppe pastoralist ancestry replaced 90% of the genetic ancestry of individuals from Britain. Our estimates for the time of gene flow in Bell Beakers samples from England suggest that the gene flow occurred ~2700 BCE (2770–2550 BCE). Our estimated dates of admixture are older than the dates of arrival of this ancestry in Britain (*Olalde et al., 2018*) and, interestingly, overlap the dates in central Europe. Given that a significant fraction of the Beaker individuals were recent migrants from central Europe, we interpret our dates reflect the admixture into ancestors of the British Beaker people, occurring in mainland Europe (*Olalde et al., 2018*).

The middle to late Bronze Age (MLBA) led to the final integration of Steppe pastoralist-related ancestry in Europe. In southern Europe, EBA samples had limited Steppe pastoralist-related ancestry, though present-day individuals harbor between ~5% and 30% of this ancestry (*Haak et al., 2015*). Using pooled samples of MLBA from Spain, we inferred major mixture occurred ~2500 BCE in Iberia. We inferred a similar timing in Italy using individuals associated with the Bell Beaker culture and EBA samples from Sicily (*Supplementary file 1B*). In Sardinia, a majority of the BA samples do not have Steppe pastoralist-related ancestry. In a few individuals, we found evidence for Steppe pastoralist-related ancestry, though in most cases, this ancestry proportion overlapped 0 and the inferred dates of admixture were very noisy (*Supplementary file 2I*). Using Iron Age samples from Sardinia, we inferred the gene flow occurred ~2600 BCE, though there is a large uncertainty associated with this estimate (3700–1490 BCE). In other parts of continental Europe and the British Isles, the Steppe pastoralist-related ancestry got diluted over time, as evidenced by more recent dates in LBA (late Bronze Age) than EBA or MBA (middle Bronze Age) samples in Germany, England, and Scotland, and an increase in Neolithic farmer ancestry during this period (*Olalde et al., 2019*; *Supplementary file 1B*).

Finally, the CWC expanded to the east to form the archaeological complexes of Sintashta, Srubnaya, Andronovo, and the BA cultures of Kazakhstan. Samples associated with these cultures harbor mixed ancestry from the Yamnaya Steppe pastoralist-related groups (CWC, in some cases) and Neolithic individuals from central Europe (*Supplementary file 2K*; *Narasimhan et al., 2019*). Applying *DATES* to eight MLBA Steppe pastoralist groups, we inferred the precise timing for the formation of these groups beginning in the third millennium BCE. These groups were formed chronologically, with the

date of genetic formation of ~3200 BCE for Sintashta culture, followed by ~2900 BCE for Srubnaya and Andronovo cultures. In the central Steppe region (present-day Kazakhstan), we obtained median dates of ~2800 BCE for the expansion of Steppe pastoralist-related ancestry in four Kazakh cultures of Maitan Alakul, Aktogai, and Kairan. By ~2700 BCE, most of these cultures had almost 60–70% Yamnaya Steppe pastoralist-related ancestry (*Supplementary file 1B*). These groups, in turn, expanded eastwards, transforming the genetic composition of populations in South Asia.

## Discussion

We developed *DATES* that measures ancestry covariance patterns in a single diploid individual genome to estimate the time of admixture. Using extensive simulations, we show that *DATES* provides accurate estimates of the timing of admixture across a range of demographic scenarios. Application of *DATES* to present-day samples shows that the results are concordant with published methods – ROLLOFF, ALDER, and Globetrotter. For sparse datasets, *DATES* outperforms published methods as it does not require phased data and works reliably with limited samples, large proportions of missing variants, as well as pseudo-haploid genotypes. This makes *DATES* ideally suited for the analysis of ancient DNA samples. We illustrate the application of *DATES* by reconstructing population movements and admixtures during the European Holocene. We confirm and extend signals that were previously identified such as the resurgence of HG ancestry during the Neolithic and provide new details about the genetic formation of the ancestral populations of Europeans and the spread of CWC and BBC cultures across Europe. Together, our analysis provides a detailed timeline and insights into the dynamics of the Neolithization of Europe and the spread of Steppe pastoralist-related ancestry across Europe.

First, we document that the Mesolithic HGs formed as a mixture of WHG and EHG ancestry ~10,200–7400 BCE. These dates are consistent with the archaeological evidence for the appearance of lithic technology associated with eastern HGs in Scandinavia and the Baltic regions (*Günther et al., 2018*; *Kashuba et al., 2019*). Next, we studied the timing of the genetic formation of Anatolian farmers. The earliest evidence of agriculture comes from the Fertile Crescent, the southern Levant, and the Zagros Mountains of Iran and dated to around 10,000 BCE. In central Anatolia, farming has been documented c. 8300 BCE (*Baird et al., 2018*; *Bellwood, 2005*). It has been long debated if Neolithic farming groups from Iran and the Levant introduced agriculture to Anatolia or HGs in the region locally adopted agricultural practices. The early Anatolian farmers can be modeled as a mixture of local HGs related to Caucasus HGs or the first farmers from Iran (*Feldman et al., 2019*). By applying *DATES* (assuming a single instantaneous admixture), we inferred that the Iran Neolithic gene flow occurred around 10,900 BCE (~12,200–9600 BCE). An alternate possibility is that there was a long period of gradual gene flow between the two groups and our dates reflect intermediate dates between the start and end of the gene flow. An upper bound for such a mixture comes from the lack of Iran Neolithic ancestry in AHGs at 13,000 BCE, and a lower bound comes from the C14 dates of early Anatolian farmers, one of which is directly dated at 8269–8210 BCE (*Feldman et al., 2019*). In either case (instantaneous admixture or gradual gene flow), the genetic mixture that formed Anatolian farmers predates the advent of agriculture in this region (*Baird et al., 2018*; *Bellwood, 2005*). This supports the model that AHGs locally transitioned to agricultural subsistence, and most probably, there was cultural diffusion from other regions in Near East (Iran and Levant) (*Feldman et al., 2019*). Future studies with more dense temporal sampling will shed light on the demographic processes that led to the transition from foraging to farming in the Near East, and in turn, elucidate the relative roles of demic and cultural diffusion in the dispersal of technologies like agriculture across populations.

Using data from 16 regions in Europe, we reconstruct a detailed chronology and dynamics of the expansion of Anatolian farmers during the Neolithic period. We infer that starting in ~6400 BCE, gene flow from Anatolian farmers became widespread across Europe, and by ~4300 BCE, it was present in almost all parts of continental Europe and the British Isles. These dates are significantly more recent than the estimates of farming based on archaeological evidence in some parts of Europe, suggesting that the local HGs and farmers coexisted for more than a millennium before the mixture occurred (*Haak et al., 2015*; *Lipson et al., 2017*). In many regions, after the initial mixture, there was a resurgence of HG ancestry, highlighting the complexities of these ancient interactions. We note that our results are consistent with two previous genetic studies, *Lipson et al., 2017*, and *Rivollat et al., 2020*, that applied genetic dating methods to a subset of samples we used in our analysis. *Lipson et al., 2017*, used a modified version of ALDER to infer the timing of admixture in three regions (*n*=151).

We obtained statistically consistent results for all overlapping samples (within two standard errors) (*Appendix 1—table 6*). An advantage of our approach over the modified ALDER approach is that we do not rely on helper samples (higher coverage individuals combined with the target group) for dating; unless these have a similar ancestry profile, they could bias the inferred dates. Our results are concordant with *Rivollat et al., 2020*, that used a previous version of *DATES* to infer the timing of Neolithic gene flow in 32 groups (vs. 86 groups in our study). We find the performance of both versions of *DATES* is similar, though some implementation details have improved (*Appendix 1—table 1*).

The second major migration occurred when populations associated with the Yamnaya culture in the Pontic-Caspian steppes expanded across Europe. Our analysis reveals the precise timing of the genetic formation of the early Steppe pastoralist groups – Yamnaya and Afanasievo – occurred ~4400–4000 BCE. This estimate predates the archaeological evidence by more than a millennium (*Anthony, 2007*) and suggests the presence of an ancient 'ghost' population of proto-Yamnaya around this time. Understanding the source and location of this ghost population will provide deep insights into the history of Pontic-Caspian cultures and the origin of Indo-European languages that have been associated to have spread with Steppe pastoralists ancestry to Europe and South Asia (*Haak et al., 2015*; *Kassian et al., 2021*). Starting in ~3200 BCE, the Yamnaya-derived cultures of CWC and BBC spread westwards, bringing Steppe pastoralist-related ancestry to Europe. Our analysis reveals striking differences in the spread of these two cultures: the CWC formation is similar across diverse regions separated by thousands of kilometers, suggesting a rapid spread after the initial formation of this group, while the spread of BBC culture was more complex and heterogeneous across regions. We find the earliest evidence of Steppe pastoralist-related ancestry in Iberia around 3200 BCE, though this ancestry only becomes widespread after 2500 BCE. In central Europe, the gene flow occurred simultaneously with archaeological evidence and was coexisting with the CWC in some parts (*Willigen and van, 2001*; *Olalde et al., 2018*). Finally, in the British Isles, the Bell Beaker culture spreads rapidly from central Europe and replaces almost 90% of the ancestry of individuals in this region (*Olalde et al., 2018*).

Recent analysis has shown remarkable parallels in the history of Europe and South Asia; with both groups deriving ancestry from local indigenous HGs, Near Eastern farmers, and Steppe pastoralist-related groups (*Narasimhan et al., 2019*). Interestingly, however, the timing of the two major migrations events differs across the two subcontinents. Both mixtures occurred in Europe almost a millennium before they occurred in South Asia. In Europe, the Neolithic migrations primarily involved Anatolian farmers, while the source of Neolithic ancestry is closer to Iran Neolithic farmers in South Asia. The Steppe pastoralist-related gene flow occurred in the context of the spread of CWC and BBC cultures in Europe around 3200–2500 BCE; in South Asia, this ancestry arrived with Steppe MLBA cultures that were formed much later in 1800–1500 BCE (*Narasimhan et al., 2019*). The Steppe MLBA groups have ancestry from Steppe pastoralist derived groups and European Neolithic farmers following the eastward expansion of CWC groups between ~3200 and 2700 BCE. Understanding the origin and migration paths of the ancestral groups thus helps to illuminate the differences in the timeline of the spread of Steppe pastoralists across the two subcontinents of Eurasia.

Genomic dating methods like *DATES* provide an independent and complementary approach for reconstructing population history. By focusing on the genetic clock based on recombination rate, we provide an independent estimate of the timing of evolutionary events up to several thousands of years. Our analysis also has advantages over the temporal sampling of ancient DNA, in that we can obtain direct estimates of when a population was formed, rather than inferring putative bounds for the timing based on the absence/presence of a particular ancestry signature (which may be sensitive to sampling choice or density). Genetic approaches provide complementary evidence to archaeology and linguistics as they date the time of admixture and not migration. Both dates are similar in many contemporary populations like African Americans and Latinos, though this may not be generally true (*Hellenthal et al., 2014*). This is underscored by our dates for the Anatolian farmer-related mixture, which postdates evidence of material culture related to agriculture by almost two millennia in some regions. This suggests that European HGs and farmers resided side by side for several thousand years before mixing (*Bollongino et al., 2013*; *Skoglund et al., 2014*). This underscores how genetic dates can provide complementary evidence to archaeology and help to build a comprehensive picture of population origins and movements.

## Materials and methods

### Dataset

We analyzed 1096 ancient European samples from 152 groups restricting to data from 1,233,013 autosomal SNP positions that were genotyped using the Affymetrix Human Origins array (the V44.3 release of the AADR; https://reich.hms.harvard.edu/allen-ancient-dna-resource-aadr-downloadable-genotypes-present-day-and-ancient-dna-data). We filtered this dataset to remove samples that were marked as contaminated, low coverage, outliers, duplicates, or first- or second-degree relatives. We grouped individuals together from a particular culture or region. Details of sample affiliation and grouping used is described in *Supplementary file 1A*.

### Modeling admixture history

We applied *qpAdm* from ADMIXTOOLS to identify the best fitting model and estimate the ancestry proportions in a target population modeled as a mixture of *n* 'reference' populations using a set of 'Outgroup' populations (*Haak et al., 2015*). We set the details: YES parameter, which reports a normally distributed Z-score to evaluate the goodness of fit of the model (standard errors were estimated with a Block Jackknife). For each target population, we chose the most parsimonious model, that is, fitting the data with the minimum number of source populations. We excluded models where the p-value < 0.05 indicating a poor fit to the data. Details of the *qpAdm* analysis for each group are reported in *Supplementary file 2*. We also applied *D*-statistics in some cases using *qpDstat* in ADMIXTOOLS with default parameters.

### *DATES*: model and implementation

*DATES* leverages the weighted ancestry covariance patterns across the genome of an admixed individual to infer the time of admixture. This method extends the idea introduced in ROLLOFF and ALDER and *Moorjani et al., 2016* to be applicable to dating admixture events between modern human populations using a single genome.

#### Basic model and notation

Assume we have an admixed individual *C* with ancestry from source populations *A* and *B*, with ancestry proportion of $\alpha$ and $\beta = (1 - \alpha)$, respectively. This mixture occurred *t* generations ago. First, we model the genotypes of *C* as a linear mix of allele frequencies of populations *A* and *B*. For any SNP *i*, let the genotype of *C* be $g_i$ and allele frequency in *A* and *B* be $p_A(i)$ and $p_B(i)$. We can then infer the mixing fraction $\alpha$ from population *A* by solving the simple linear regression by minimizing the residuals.

$$R = \sum_i \left( g_i - \left( \alpha p_A(i) + (1 - \alpha) p_B(i) \right) \right)^2 \tag{1}$$

Let $a_i$ be the probability of observing $g_i$ in *C* given the observed genotype in *A*, and $b_i$ be the probability of observing $g_i$ in *C* given the observed genotype in *B*

$$a_i = P(g_i|A)$$
$$b_i = P(g_i|B)$$

We can then compute the likelihood $L_i$ of observing a genotype $g_i$ in the admixed individual

$$L_i = \alpha a_i + \beta b_i \tag{2}$$

For a pair of neighboring markers $S_1$, $S_2$ located at a genetic distance of *d* Morgans, the probability of no recombination between the two markers is given by $\theta = e^{-td}$. Accounting for recombination, the log-likelihood that the two markers have the same ancestry is then given by:

$$\mathcal{L} = \log \left[ (1 - \theta) L_1 L_2 + \theta \left( \alpha a_1 a_2 + \beta b_1 b_2 \right) \right] \tag{3}$$

Let $K_i$ represent the ancestry at marker $S_i$. Expanding as a power series in $\theta$, the coefficient of $\theta$ is $QK_1K_2$, where

$$Q = \alpha\beta$$

$$K_i = \frac{(a_i - b_i)}{L_i} \tag{4}$$

We can compute the ancestry covariance, $A(d)$, across pairs of markers $S_1$, $S_2$ separated by distance $d$ as

$$A(d) = \frac{\sum_{s(d)} (K_1 - \bar{K}_1)(K_2 - \bar{K}_2)}{|S(d)|}$$

where $S(d)$ is a set of markers $S_1$, $S_2$ located $d$ Morgans apart.

The ancestry covariance $A(d)$ is expected to follow an exponential decay with $d$ with the rate of decay depending on the time since admixture ($t+1$).

$$A(d) \sim e^{-(t+1)d}$$

The factor of ($t+1$) comes from the insight that in the first generation following admixture, the admixed population derives one chromosome from each ancestral group. The mixing of chromosomes only begins in the following generations as the chromosomes recombine. This means that if we fit $t$ generations, we are likely to underestimate the time of admixture. We note that previous methods like ALDER and ROLLOFF, however, incorrectly fit $t$ generations to infer the time of mixture. In practice, however, this has little effect on the inference except maybe in case of very recent admixture dates. We infer the time of the mixture by fitting an exponential distribution with affine term using least squares. *DATES* is applicable for dating admixture in a single individual. When multiple individuals from an admixed population are available, *DATES* computes the log-likelihood by summing over all individuals.

## Application to real data

We applied *DATES* using genome-wide SNP data from the target population and two reference populations. To infer the allele frequency in the ancestral populations more reliably, where specified, we pooled individuals deriving the majority of their ancestry from the population of interest (**Supplementary file 1A**). We computed the weighted ancestry covariance between 0.45 cM (centiMorgans) (to minimize the impact of background LD) and 100 cM, with a bin size of 0.1 cM. We plotted the weighted covariance with genetic distance and obtained a date by fitting an exponential function with an affine term $y = Ae^{-\lambda d} + c$, where $d$ is the genetic distance in Morgans and $\lambda = (t+1)$ is the number of generations since admixture ($t$). We computed standard errors using weighted block jackknife, where one chromosome was removed in each run (**Busing et al., 1999**). Following **Tournebize et al., 2020**, we examined the quality of the exponential fit by computing the normalized root-mean-square deviation (NRMSD) between the empirical ancestry covariance values $z$ and the fitted ones $\hat{z}$, across all the genetic distance bins (Appendix 1).

$$NRMSD = \frac{1}{max(\hat{z}) - min(\hat{z})} \sqrt{\frac{\sum^D (z - \hat{z})^2}{N}}$$

The estimated dates of admixture were considered significant if the (a) $Z$-score > 2, (b) $\lambda < 200$ generations and (c) NRMSD < 0.7. We converted the inferred dates from generations to years by assuming a mean generation time of 28 years (**Moorjani et al., 2016**). For ancient samples, we added the sampling age of the ancient specimen (**Supplementary file 1A**). When multiple individuals were available, we used the average sampling ages to offset the admixture dates. We report dates in BCE by assuming the 1950 convention.

## Comparison of old and new version of *DATES*

An earlier version of *DATES* (version v753) was released in **Narasimhan et al., 2019**. The current method (version 4010) released in this study differs in some key aspects of the implementation as described below.

a.  Use of regression model vs. likelihood approach: In v753, we used a regression model to infer the residuals at each site in the genotype by conditioning on the allele frequency in the reference population and the genome-wide estimate of the admixture proportion (**Narasimhan**

*et al., 2019*). In contrast, in the current version (v4010), we use a more rigorous likelihood framework where we infer the probability of ancestry from each reference population at each site in the genome (*Equation 3*).

b. Rate of decay of exponential fit: In v753, like ALDER and ROLLOFF, we fit an exponential decay with the rate of *t* generations. However, this assumes that mosaic chromosomes are formed in the generation when the gene flow occurs. However, in reality, the mixing of ancestry only begins in the following generations as the chromosomes of distinct ancestry recombine. To correctly account for this effect, we fit an exponential with the rate of (*t*+1) in *DATES* v4010. In practice, this has a minor effect on the dates reported earlier, as in most cases the uncertainty is much larger than one generation.

c. Goodness of fit test: In v4010, we implemented the NRMSD to assess the fit of the exponential curve. NRMSD computes the deviation between the empirical estimate and fitted data in order to provide a statistical way to characterize the noisiness of the fitted curve. Lower values of NRMSD suggest a better fit, however, there is no clear interpretation of the absolute value of NRMSD. Based on the empirical distribution of NRMSD values in our study samples (*Appendix 1—figure 3*), we infer a conservative threshold of 0.7 to define a 'good' fit. We caution that users should adjust this threshold based on their application and always visually inspect their exponential fits to ensure reliable results.

d. Support for arbitrary number of chromosomes: Unlike v753 that was optimized for parameters in humans, the new version supports an arbitrary number of chromosomes (inputted by the user) so *DATES* can be used in any species.

A comparison of the two version of *DATES* using simulated data (*Appendix 1—table 1*) and empirical data (*Appendix 1—table 2*, *Appendix 1—table 3*) yields qualitatively similar results.

## Simulations

We constructed admixed genomes following the approach described in *Moorjani et al., 2011*. This method requires phased haplotypes from two source populations and uses two key parameters to simulate data from admixed individuals, (a) the mixture proportion ($\alpha$) that represents the probability that a particular sampled haplotype comes from one of the reference panels, namely *source₁* and *source₂*, and (b) the time of mixture ($\lambda$) which is the number of generations since mixture. To simulate an admixed individual, we begin at the start of the chromosome and sample a haplotype from either *source₁* with a probability ($\alpha$) and *source₂* with a probability ($1 - \alpha$). At each subsequent marker, we check if there was a recombination event between the two neighboring markers. A recombination event occurs with a probability of ($1 - e^{-\lambda g}$), where *g* is the genetic distance in Morgans. We use the time of $\lambda = (t + 1)$ generations to account for the fact that in the first generation following admixture, the offspring inherits one chromosome of each ancestry. In the next generation, the crossovers lead to a mixing of ancestry. Thus, when a recombination event occurs, we resample the ancestry between *source₁* and *source₂*, otherwise, we copy the haplotype from the same source population. (Note, a recombination event can lead to a switch to a haplotype of the same ancestry.) Once the ancestry is chosen, we randomly pick a haplotype from the ancestral pool (without replacement) and copy its sequence to the genome of the admixed individual. This process is continued until we reach the end of the chromosome. Using this approach, we generate the genomes of *n* admixed individuals. The simulated haploid chromosomes are merged at random to construct diploid admixed individuals. This algorithm requires more than 2*n* ancestral haplotypes for generating data for *n* diploid admixed individuals (*Moorjani et al., 2011*). For more than two reference populations, the same algorithm is repeated iteratively.

We used 111 CEU and 112 YRI phased 1000 genomes phase 3 dataset (*Auton et al., 2015*) for generating 10 admixed genomes for ~380,000 SNPs (unless otherwise stated). For the inference, we used French and Yoruba from HGDP (*Li et al., 2008*). We generated data for various demographic scenarios, where we varied the time the admixture ($\lambda$), proportion of mixture ($\alpha$), sample size in the reference and target populations, divergence between the ancestral and reference populations used and studied their impact on the estimated dates. We also characterized the impact of features of ancient DNA such as missing data, pseudo-haploid genotypes, and limited sample size. In order to simulate pseudo-haploid genotypes, we randomly sampled an allele at each heterozygous site and assigned it as the homozygous genotype at that site (*Harney et al., 2021*). To generate missing

data, we set the genotype call at a site as 'missing' or 'unknown' (in eigenstrat format as 9) where the proportion of missing genotypes ranged between 5% and 60% in our simulations. We also evaluated the impact of the choice of reference populations used in *DATES* in case of simple and multiple pulses of admixture.

To study the impact of complex scenarios of admixture involving founder events and continuous gene flow, we used a coalescent simulator, MaCs (*Chen et al., 2009*). We simulated 100 Mb of three populations with an effective population size of 12,500, mutation rate of $1.2 \times 10^{-8}$ and recombination rate $1 \times 10^{-8}$ per base pair per generation, respectively (*Halldorsson et al., 2019*; *Jónsson et al., 2017*). We assumed the admixture occurred continuously over a period of time or was followed by the bottleneck. In case of the latter, the duration of the bottleneck was 1–10 generations with reduction in effective population size from 12,500 to 10–1000 and the population recovered to its original size after the bottleneck or maintained a small size until present (no recovery founder event). For each simulation, we generated data for two haploid chromosomes and combined these to generate one diploid chromosome.

## Software availability

The executable and source code for *DATES* will be available on GitHub: https://github.com/Moor-janiLab/DATES_v4010 (copy archived at swh:1:rev:e034dc0d6fe8d41a828796f07791d50011b6bb04; *Chintalapati et al., 2022*).

## Acknowledgements

We thank Monty Slatkin, Ziyue Gao, David Reich, Iosif Lazaridis, and Vagheesh Narasimhan for their comments on the manuscript. We thank Iosif Lazaridis for helpful discussions about population models in the Near East, Remi Tournebize for suggestions for evaluating the fit of exponential decay curves, and Neel Alex for suggestions for implementation of FFT for an earlier version of DATES.

---

## Additional information

### Funding

| Funder | Grant reference number | Author |
|---|---|---|
| National Institutes of Health | R35GM142978 | Priya Moorjani |
| Burroughs Wellcome Fund | Career Award at the Scientific Interface | Priya Moorjani |
| Alfred P. Sloan Foundation | Sloan Research Fellowship | Priya Moorjani |

The funders had no role in study design, data collection and interpretation, or the decision to submit the work for publication.

### Author contributions

Manjusha Chintalapati, Conceptualization, Data curation, Formal analysis, Investigation, Methodology, Validation, Visualization, Writing – original draft, Writing – review and editing; Nick Patterson, Software, Writing – review and editing; Priya Moorjani, Conceptualization, Formal analysis, Funding acquisition, Investigation, Methodology, Project administration, Supervision, Validation, Writing – original draft, Writing – review and editing

### Author ORCIDs

Manjusha Chintalapati  http://orcid.org/0000-0002-9808-5991
Nick Patterson  http://orcid.org/0000-0003-0302-684X
Priya Moorjani  http://orcid.org/0000-0002-0947-5673

### Decision letter and Author response

Decision letter https://doi.org/10.7554/eLife.77625.sa1
Author response https://doi.org/10.7554/eLife.77625.sa2

# Additional files

## Supplementary files

• Supplementary file 1. Data and admixture dates inferred using *DATES (Distribution of Ancestry Tracts of Evolutionary Signals)* for European groups during the Holocene (Excel sheet). (A) Information on ancient samples used in our study. (B) Estimated dates of admixture for population mixture events during the European Holocene.

• Supplementary file 2. Formal tests of admixture for populations in Europe using *qpAdm* and *D*-statistics with default parameters in ADMIXTOOLS (Excel sheet). (A) Modeling population admixture of hunter-gatherer (HG) groups using *qpAdm* in ADMIXTOOLS. (B) *D*-statistics to assess the affinity of Mesolithic HG groups to western hunter-gatherers (WHGs) or Anatolian farmers. (C) Modeling population admixture of Near Eastern farmers using *qpAdm* in ADMIXTOOLS. (D) Modeling population admixture of Neolithic European groups using *qpAdm* in ADMIXTOOLS. (E) Modeling population admixture of Neolithic European groups per individual using *qpAdm* in ADMIXTOOLS. (F) *D*-statistics to explore the affinity of the target groups to Steppe pastoralists or Anatolian farmers. (G) Modeling population admixture of Early Steppe pastoralists groups using *qpAdm* in ADMIXTOOLS. (H) Genetic distance (FST) in early Steppe pastoralists groups. (I) Modeling population admixture of Bronze Age groups using *qpAdm* in ADMIXTOOLS. (J) Modeling population admixture of Bronze Age groups per individual using *qpAdm* in ADMIXTOOLS. (K) Modeling population admixture of middle to late Bronze Age (MLBA) Steppe pastoralists groups. Age groups using *qpAdm* in ADMIXTOOLS.

• Transparent reporting form

## Data availability

All data analyzed during this study is publicly available at: https://reich.hms.harvard.edu/allen-ancient-dna-resource-aadr-downloadable-genotypes-present-day-and-ancient-dna-data. R code to replicate figures and figure supplements is available at https://github.com/manjushachintalapati/DATES_EuropeanHolocene/ (copy archived at swh:1:rev:041ccbed941eb0ca188b188e892fb7f89478f871).

The following previously published dataset was used:

| Author(s) | Year | Dataset title | Dataset URL | Database and Identifier |
|-----------|------|---------------|-------------|-------------------------|
| David Reich Lab | 2021 | Allen Ancient DNA Resource | https://reich.hms.harvard.edu/allen-ancient-dna-resource-aadr-downloadable-genotypes-present-day-and-ancient-dna-data | Harvard University, V44.3 |

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

## Appendix 1

### *DATES*: implementation, versions, and comparison with other published methods

#### *DATES* FFT implementation

In order to make *DATES* computationally tractable, we implemented the FFT for computing ancestry covariance as described in ALDER (*Loh et al., 2013*). Briefly, we perform an algebraic transformation of the ancestry covariance statistic (described below) and compute the FFT convolution in discrete equally sized bins (referred to as mesh points).

In *DATES*, we compute the ancestry covariance $A\left(d\right)$ (Materials and methods) by expanding the numerator as below. $X_j\left(d\right)$, ($j$=0, 1, 2), where

$$X_2\left(d\right) = \sum_{s(d)} K_1 K_2$$

$$X_1\left(d\right) = 2\left(\sum_{s(d)} \bar{K}_1 K_2 + \bar{K}_2 K_1\right)$$

$$X_0\left(d\right) = \sum_{s(d)} \bar{K}_1\, \bar{K}_2$$

where $d$ is genetic distance in Morgans between pair of neighboring markers $S_1$, $S_2$ and $K_i$ represents the ancestry at marker $S_i$. We discuss an approximate calculation of $X_2$. The calculations of $X_1$ and $X_0$ are similar.

Like ALDER, we divide the genome in windows based on the position in the genetic map (instead of genetic distance). We set a mesh on the genetic map (default mesh size is 0.01 cM), mapping every SNP to the nearest mesh point. For a mesh point, $u$ define $T_u$ to be the set of SNPs mapping to $u$ and

$$K\left(u\right) = \sum_{i \in T(u)} K_i$$

We now set

$$X_2'\left(d\right) = \sum_{u,v:\ |u-v|=d} K\left(u\right) K\left(v\right)$$

where $|u-v|$ is the genetic distance of $u, v$. $X_2'$ can be computed by FFT. We note that the use of the mesh is the only source of approximation in the FFT implementation to compute $X_2$. The mesh discretization parameter, *qbin*, provides a trade-off between runtime and accuracy, smaller mesh size leads to higher accuracy and longer runtime.

To explore the impact of *qbin* on the estimated accuracy, we performed simulations for varying sample sizes (*n*=1 and *n*=20) and ran *DATES* using varying *qbin* between 1 and 100. We find the method works reliably for all *qbin* values (*Appendix 1—figure 1*). Moreover, there is almost a 5- to 10-fold speedup in a run between *qbin* values of 10 vs. 100 (*Appendix 1—figure 2*). The runtime is invariant to the proportion or time of admixture. The default value of *qbin* in *DATES* is 10 but we advise the user to perform simulations for their dataset size and population model to set this parameter reliably.

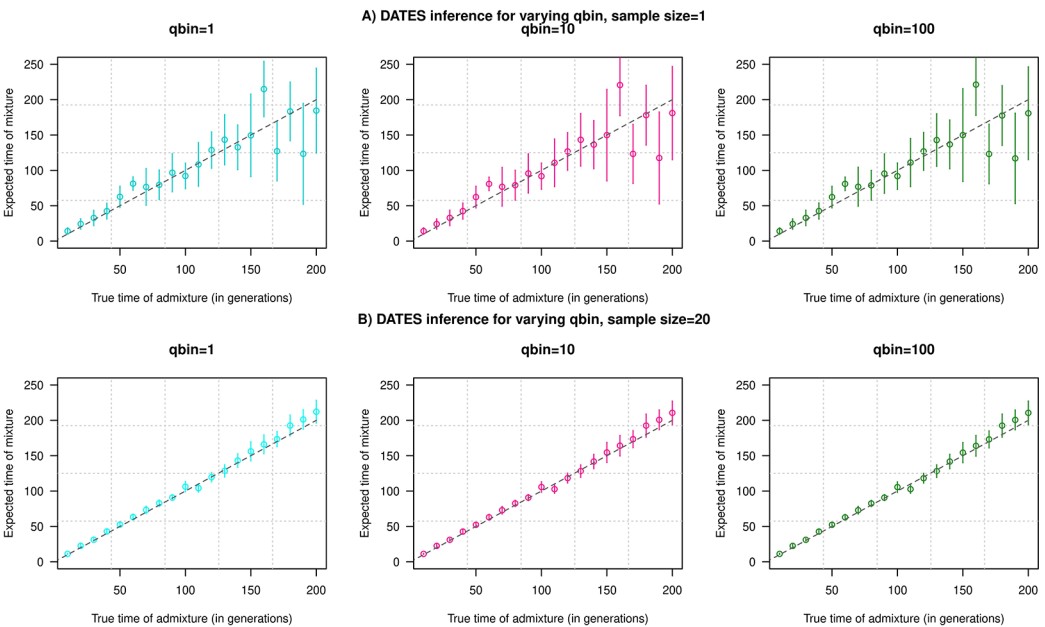

**Appendix 1—figure 1.** Impact of the discretization parameter (*qbin*) on accuracy. We show three subplots for a sample size of *n*=1 (Panel A) and *n*=20 (Panel B). For each subplot, we simulated data for *n* admixed individuals with 20% ancestry from Europeans (1000 Genomes, CEU) and 80% ancestry from Africans (1000 Genomes, YRI) with the time of admixture ($\lambda$) shown on the X-axis and the estimated admixture time inferred using *DATES* on Y-axis. We ran *DATES* using varying *qbin* values shown in different colors.

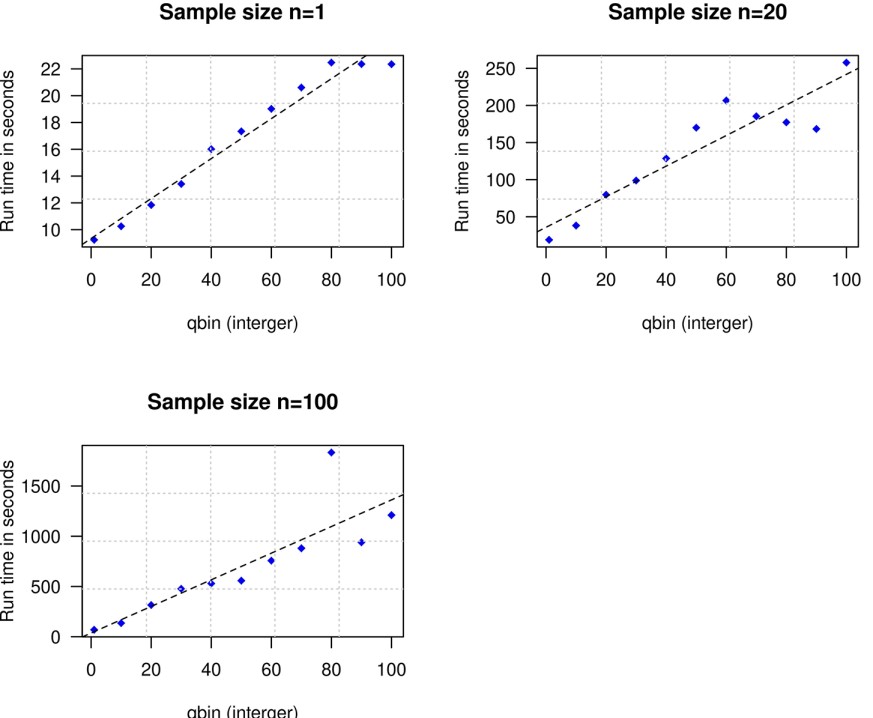

**Appendix 1—figure 2.** Impact of the discretization parameter (*qbin*) on runtime. We show three subplots for sample size of *n*=1 (top left), *n*=20 (top right), and *n*=100 (bottom). For each subplot, we simulated data for *n* admixed individuals with 20% ancestry from Europeans (1000 Genomes, CEU) and 80% ancestry from Africans (1000 Genomes, YRI) with the time of admixture ($\lambda$) of 100 generations ago. We show the impact of *qbin* (X-axis) on the runtime measured in seconds. For sample sizes, *n*>1, $r^2$ between *qbin* and runtime is >0.99.

## Assessing the exponential fit

Following *Tournebize et al., 2020*, we examined the quality of the exponential fit by computing the normalized root-mean-square deviation (NRMSD) between the empirical ancestry covariance values $z$ and the fitted ones $\hat{z}$, across all the genetic distance bins (where $N$ is the number of bins) (*Tournebize et al., 2020*).

$$NRMSD = \frac{1}{max(\hat{z}) - min(\hat{z})} \sqrt{\frac{\sum^{D}(z-\hat{z})^2}{N}}$$

We calculated NRMSD for all ancient DNA populations in our study and the distribution of these values is shown in *Appendix 1—figure 3*. Focusing on the most extreme values of NRMSD, we show that the statistic is useful in identifying poor fits where the fitted line deviates from the data or the fit is highly dispersed (*Appendix 1—figure 4*). However, the absolute value of this statistic does not have any statistical meaning. Based on the empirical distribution of NRMSD values in our study samples (*Appendix 1—figure 3*), we use a threshold of 0.7 to flag poor fits. We caution that users should adjust this threshold based on their application and always visually inspect their exponential fits to ensure reliable results.

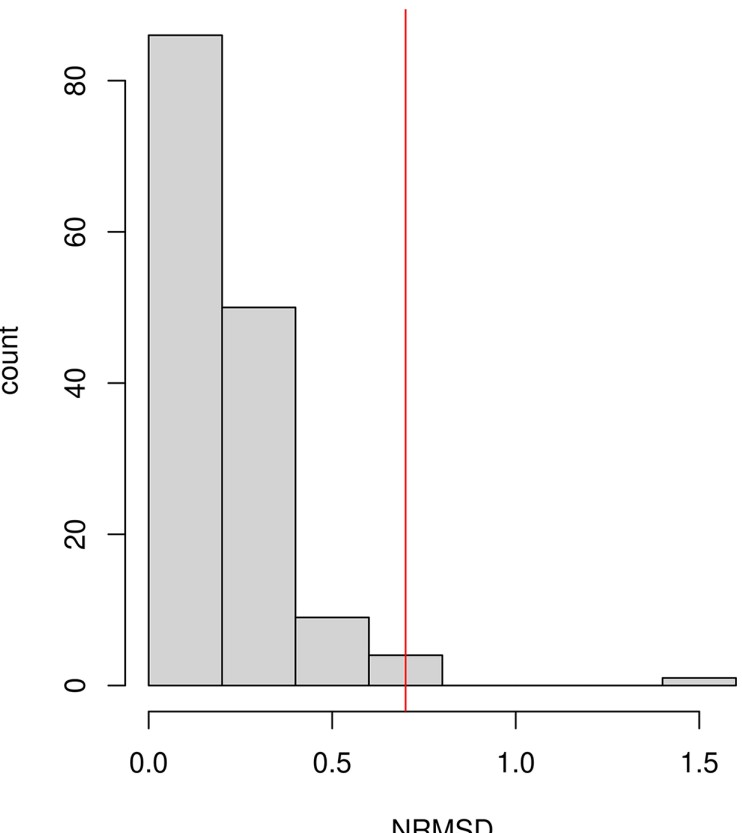

**Appendix 1—figure 3.** Histogram of the normalized root-mean-square deviation (NRMSD) values computed as the normalized residual between the empirical and fitted decay curves, for all the ancient DNA populations reported in *Figure 2—figure supplement 1*. The red vertical line represents the value NRMSD = 0.7, which we used as the threshold to exclude populations from our analysis because visual inspection of fitted curves above this threshold suggests the results are too noisy to make a reliable inference (see *Figure 2—figure supplement 1* for all fitted decay curves).

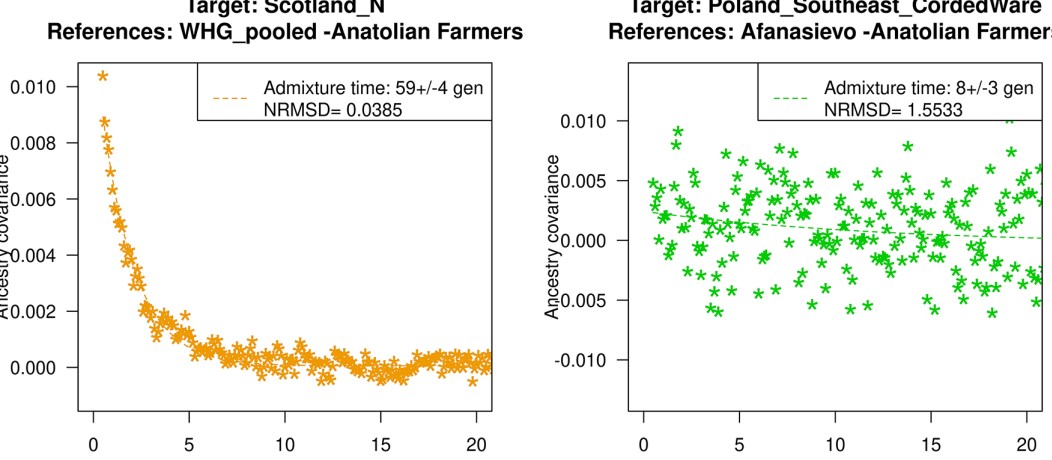

**Appendix 1—figure 4.** Ancestry covariance curves for the lowest (left) and highest (right) NRMSD values in our ancient DNA populations in our study. For details of all curves and NRMSD estimation, see *Figure 2—figure supplement 1*.

## Comparison of versions of *DATES*

An earlier version of *DATES* (version v753) was released in *Narasimhan et al., 2019*. The current method (version 4010) released in this study differs in some key aspects of the implementation (Materials and methods). To compare the two versions of *DATES*, we performed simulations and generated 10 admixed individuals with 20% European and 80% African ancestry where the time of admixture varied between 10 and 300 generations (similar to *Appendix 2—figure 1*). We also varied the sample sizes of the admixed population between 1 and 20 in increments of 5. Our estimated admixture using v753 and v4010 are highly concordant suggesting although the implementation has changed, the results are similar (*Appendix 1—table 1A–B*). Further, we compared the dates of times of admixture that were reported using the earlier version. To this end, we repeated the analysis for *Narasimhan et al., 2019*, for ancient South Asians and *Rivollat et al., 2020* for ancient Neolithic samples in Europe (*Rivollat et al., 2020*). In both cases, we obtained consistent results as reported earlier (*Appendix 1—table 2*, *Appendix 1—table 3*).

**Appendix 1—table 1.** Comparison of results using *DATES (Distribution of Ancestry Tracts of Evolutionary Signals)* v753 and v4010 using simulated data.

**(A) Simulated data with the target sample size (*n*) of 10 individuals**

| True time of admixture (generations) | *DATES* (V753) (mean ± SE) | *DATES* (v4010) (mean ± SE) |
|---|---|---|
| 10 | 10.0±0.5 | 11.0±0.6 |
| 20 | 19.1±1.5 | 19.6±1.5 |
| 30 | 28.0±1.5 | 28.9±1.3 |
| 40 | 46.1±2.1 | 45.7±1.8 |
| 50 | 55.5±2.9 | 55.6±2.8 |
| 60 | 59.4±2.2 | 60.5±2.4 |
| 70 | 69.3±4.2 | 69.8±4.0 |
| 80 | 84.2±4.4 | 84.0±3.9 |
| 90 | 97.7±3.7 | 93.7±3.6 |
| 100 | 107.4±5.4 | 106.7±4.5 |
| 110 | 113.6±5.5 | 112.9±4.7 |
| 120 | 122.7±5.4 | 124.3±5.5 |

*Appendix 1—table 1 Continued on next page*

*Appendix 1—table 1 Continued*

**(A) Simulated data with the target sample size (*n*) of 10 individuals**

| True time of admixture (generations) | DATES (V753) (mean ± SE) | DATES (v4010) (mean ± SE) |
|---|---|---|
| 130 | 138.6±7.7 | 134.1±6.2 |
| 140 | 153.2±9.2 | 152.0±8.4 |
| 150 | 147.5±9.0 | 146.6±8.2 |
| 160 | 181.4±9.8 | 176.6±8.2 |
| 170 | 178.0±8.1 | 175.9±7.4 |
| 180 | 180.7±10.6 | 182.3±9.3 |
| 190 | 172.9±15.3 | 174.8±12.7 |
| 200 | 204.7±17.1 | 208.8±13.4 |
| 210 | 194.5±13.3 | 196.3±11.9 |
| 220 | 255.8±17.0 | 250.7±13.8 |
| 230 | 251.9±18.6 | 237.0±13.0 |
| 240 | 234.7±18.3 | 241.5±14.2 |
| 250 | 228.3±13.8 | 233.2±11.9 |
| 260 | 254.2±21.7 | 253.0±16.1 |
| 270 | 291.4±22.4 | 292.1±20.0 |
| 280 | 252.4±25.2 | 248.1±22.4 |
| 290 | 277.9±22.4 | 285.4±20.5 |
| 300 | 318.6±23.3 | 315.1±20.3 |

(B) Simulated data with sample size (n) ranging between 10 and 20 individuals.

| True time of admixture (generations) | Sample size | DATES (V753) (mean ±SE) | DATES (v4010) (mean ±SE) |
|---|---|---|---|
| 10 | 1 | 6.9±2.5 | 7.8±2.6 |
| 10 | 5 | 8.8±0.8 | 9.9±0.8 |
| 10 | 10 | 9.9±1.2 | 10.8±1.2 |
| 10 | 15 | 10.9±0.7 | 11.8±0.7 |
| 10 | 20 | 10.3±0.6 | 11.3±0.6 |
| 50 | 1 | 51.7±7.1 | 51.5±7.5 |
| 50 | 5 | 59.7±3.5 | 58.6±3.2 |
| 50 | 10 | 48.7±2.7 | 50.1±2.7 |
| 50 | 15 | 54.2±2.1 | 54.7±2.1 |
| 50 | 20 | 52.9±1.9 | 53.1±1.9 |
| 100 | 1 | 124.5±17.6 | 122.5±13.2 |
| 100 | 5 | 107.2±7.6 | 108.2±7.5 |
| 100 | 10 | 103.3±7.7 | 100.1±3.8 |
| 100 | 15 | 99.4±4.5 | 103.4±3.3 |
| 150 | 1 | 136.4±29.2 | 144.2±26.3 |
| 150 | 5 | 142.6±11.4 | 143±11.7 |
| 150 | 10 | 156.9±9 | 158.6±7 |
| 150 | 15 | 142.9±7.8 | 146.1±6.7 |
| 150 | 20 | 156.5±5.4 | 152.9±4.3 |
| 200 | 1 | 195.4±88.4 | 160.2±73.9 |

*Continued on next page*

*Continued*

(B) Simulated data with sample size (n) ranging between 10 and 20 individuals.

| True time of admixture (generations) | Sample size | DATES (V753) (mean ±SE) | DATES (v4010) (mean ±SE) |
|---|---|---|---|
| 200 | 5 | 210.9±20.7 | 206.7±18.7 |
| 200 | 10 | 225±18.7 | 219.8±18 |
| 200 | 15 | 200±10.6 | 197.7±9 |
| 200 | 20 | 189.4±11 | 190.3±9 |

**Appendix 1—table 2.** Comparison of results with *Narasimhan et al., 2019*.

| Population | Reference populations* | DATES (v753) | DATES (v4010) |
|---|---|---|---|
| | | (mean ±SE; in generations) | (mean ±SE; in generations) |
| *Indus_Periphery_Pool* | AASI and Iranian-farmer-related | 71±15 | 62±7 |
| *SPGT* | AASI and Steppe-pastoralist-related | 26±3 | 28±3 |

*We used the reference populations of AASI ancestry that includes South Asians from the 1000 Genomes Project (Phase 3) including Sri Lankan Tamil from the UK (*STU.SG*) and Indian Telugu from the UK (*ITU.SG*), as well as *BIR.SG*, and Iranian farmer-related ancestry including Aigyrzhal_BA, Sarazm_EN, Geoksyur_EN, Parkhai_Anau_EN, and Steppe-pastoralist-related including Central_Steppe_MLBA.

**Appendix 1—table 3.** Comparison of dates of the spread of Neolithic farming from *Rivollat et al., 2020*.

| Population | n | DATES (v753) | Population in our study (v44 1240K) | n (v44 1240K) | DATES (v4010)[#] |
|---|---|---|---|---|---|
| Bulgaria_MP_Neolithic | 9 | 8.4±2.3 | Bulgaria_MalakPreslavets_N | 3 | 8.05±3 |
| Serbia_Neolithic | 4 | – | Serbia_EN | 3 | 22.8±9.8 |
| Romania_EN | 2 | 32.1±10.4 | Romania_EN* | 2 | 29.7±7.1 |
| Croatia_Impressa | 2 | – | Croatia_EN_Impressa | 2 | – |
| Hungary_ALPc_MN | 23 | 21.5±4.7 | Hungary_MN_ALPc | 21 | 21.9±1.6 |
| Hungary_LBK_MN | 10 | 12.8±5.2 | Hungary_MN_LBK | 6 | 18.6±7.4 |
| Hungary_ALBK_MN | 2 | 14.8±3.2 | Hungary_MN_ALBK_Szakalhat | 2 | 19.3±3.3 |
| Hungary_LN | 18 | 21.5±3.7 | Hungary_LN | 18 | 28.03±3.8 |
| Austria_LBK_EN | 8 | 15.5±4.6 | Austria_EN_LBK | 9 | 17.6±2.3 |
| Czech_MN | 5 | 18.3±7.7 | Czech_MN | 4 | 32.9±6.3 |
| France_MN | 3 | 26.5±5.6 | France_MN | 43 | 30±1.3 |
| Iberia_EN | 10 | 15.6±2.5 | Spain_EN | 11 | 20.6±3.6 |
| Iberia_MN | 7 | 52.4±4.3 | Spain_MLN | 42 | 56.3±4 |
| Germany_LBK_EN | 27 | 14.4±2.6 | Germany_EN_LBK | 54 | 17.4±2.7 |
| Germany_Blatterhohle_MN | 4 | 12.3±2.5 | Germany_Blatterhohle_MN | 4 | 16.2±2.9 |
| Germany_Esperstedt_MN | 1 | – | Germany_MN_Esperstedt | 1 | – |
| England_Neolithic | 29 | 45.5±5.5 | England_N.SG | 17 | – |
| Wales_Neolithic | 6 | 45.3±7.4 | Wales_N | 4 | 50.7±3.3 |
| Scotland_Neolithic | 42 | 50.9±3.8 | Scotland_N | 30 | 56.6±2.9 |
| Ireland_Neolithic | 13 | 46.9±7.5 | Ireland_MN.SG | 26 | 50.8±2.2 |

(Blue) indicates samples sizes that differ across both studies.

# For DATES (Distribution of Ancestry Tracts of Evolutionary Signals), we used pooled western hunter-gatherer (WHG) and Anatolian farmers as the reference populations except for samples marked with *.

*Indicates cases where the results were not significant as the 95% CI includes 0.

## Comparison of DATES with published methods
### Comparing DATES and ALDER in simulations

To compare the *DATES* with ALDER, we simulated data for *n* (=20 or 100) admixed individuals with 20% European and 80% African ancestry using CEU and YRI phased individuals from 1000 Genomes Project (*Auton et al., 2015*). We used French and Yoruba from the HGDP dataset (*Li et al., 2008*) as the reference populations to represent European and African source populations, respectively. We applied *DATES* and ALDER to the same dataset. We ran ALDER using default settings and allowed ALDER to pick the minimum distance to start the exponential fit. ALDER estimates the date of admixture by fitting an exponential to the weighted covariance statistic with genetic distance and performs a least-squares fit using $y = Ae^{-td} + c$, where $d$ is the genetic distance in Morgans and $t$ is the number of generations since admixture. We note this differs from *DATES* which assumes the exponential decay parameter of $(t + 1)$, though in practice this has little effect on the comparisons. For the timing of admixture between 10 and 200 generations, we observed that both methods accurately estimated the time of admixture in most cases, though *DATES* provided more precise estimates than ALDER for older admixture dates (*Appendix 1—table 4*).

**Appendix 1—table 4.** Comparison of ALDER and *DATES (Distribution of Ancestry Tracts of Evolutionary Signals)* for varying samples sizes and times of admixture.

We simulated data for 20 and 100 admixed individuals using the CEU and YRI from 1000G with the mixture proportion of 20% from European and 80% African ancestry. The dates reported here for *DATES* are using exponential fit to $\lambda - 1$ generations.

| | Number of individuals, *n*=20 | | Number of individuals, *n*=100 | |
| Time of admixture (gen) | ALDER mean ±1 SE (gen) | *DATES* mean ±1SE (gen) | ALDER mean ±1SE (gen) | *DATES* mean ±1SE (gen) |
| --- | --- | --- | --- | --- |
| 10 | 9.3±0.8 | 10.7±0.6 | 10.2±0.3 | 10±0.3 |
| 20 | 19.4±1.3 | 19.7±0.8 | 20.2±0.3 | 20.3±0.3 |
| 30 | 28.5±1.7 | 30.8±1.5 | 30.6±0.9 | 30.5±0.7 |
| 40 | 40.9±2 | 40.3±1.5 | 40.6±0.7 | 40.6±0.4 |
| 50 | 47.9±3.6 | 49.6±1.6 | 50±1.1 | 50.9±0.7 |
| 60 | 55.7±2.7 | 60.3±1.5 | 62±2.2 | 63.2±1 |
| 70 | 71.4±4 | 74±2.7 | 74±2.2 | 72.4±1.3 |
| 80 | 80.6±4.8 | 82.5±2.9 | 85.3±2.3 | 84.4±1.1 |
| 90 | 87.8±4.2 | 88.9±3 | 94.1±2.7 | 92.9±1.3 |
| 100 | 93.7±4.9 | 98.1±2.9 | 101.9±3.9 | 103.6±1 |
| 110 | 121.4±5.4 | 118.2±3.7 | 120.7±4.3 | 115.5±1.8 |
| 120 | 116.5±8.5 | 128.4±3.9 | 121.2±5.1 | 121.5±1.7 |
| 130 | 138.2±9.2 | 133.7±4.6 | 130.2±4.8 | 132.8±1.7 |
| 140 | 134.5±17.5 | 142.4±7 | 144.9±7.3 | 145.3±3.1 |
| 150 | 144.8±23.8 | 149.5±7.4 | 155.1±7 | 157.5±2.8 |
| 160 | 141.9±11.3 | 166.7±5.9 | 154.5±8.5 | 161.7±2.4 |
| 170 | 173.4±13.7 | 175.1±6.9 | 170.3±6.2 | 173.5±3 |
| 180 | 204.6±17.8 | 195.5±7.1 | 174.2±7 | 180.7±3.3 |
| 190 | 221.3±23.9 | 210.4±9.4 | 191.2±16.2 | 197.2±4.6 |
| 200 | 202.8±11.1 | 196±6.5 | 188.5±16.5 | 202.6±4.7 |

Next, we generated 10 simulated individuals with missing genotypes varying between 5% and 60% (in increments of 5%) as described in Materials and methods and applied both *DATES* and ALDER. Using the same setup in both methods, we inferred that *DATES* reliably recovers the time of admixture even when samples had large missing proportions of around 60% (, *Appendix 1—figure 5*). In contrast, ALDER becomes every noisy with moderate proportions of missing data (>40%). For older dates (>100 generations), ALDER estimates appear to be biased even with >10% missing genotypes (*Appendix 1—figure 5*). As the missing sites vary among individuals, admixture LD-

based methods such as ALDER that combine information across individuals become noisy as there are few sites without non-missing genotypes remaining for inference. However, *DATES* performs the analysis for single individuals (using all non-missing genotypes for that individual) and then averages the inferred estimates across individuals. This provides substantial robustness to variable missingness across individuals.

## Comparison of *DATES* with published admixture dating methods – ROLLOFF, Globetrotter, ALDER

**Appendix 1—table 5.** Admixture dates in present-day populations inferred using ROLLOFF, Globetrotter, ALDER, and *DATES (Distribution of Ancestry Tracts of Evolutionary Signals).*

| Population | nk | Source1 | Source2 | ROLLOFF | Globetrotter | ALDER formal test | ALDER_2-ref dates | *DATES* | Comments |
|---|---|---|---|---|---|---|---|---|---|
| Hazara | 22 | Mongola (10) | Iranian (13) | 23±1 | 22±0.9 | Long-range LD | -- | 24.6±1.0 | |
| Uzbekistani | 15 | Mongola (10) | Iranian (13) | 20±1.4 | 19±1.1 | SUCCEEDS | 19.18±2.22 | 21.3±1.4 | |
| Uyghur | 10 | Mongola (10) | Iranian (13) | 23±2.6 | 22±1.3 | SUCCEEDS | 16.73±1.38 | 22.2±2.1 | |
| Makrani | 22 | Bantu Kenya (11) | Balochi (21) | 18±1.8 | 18±1.2 | Long-range LD | -- | 13.2±1.6 | |
| Druze | 42 | Yoruba (21) | Cypriot (12) | 39±7.3 | 37±1.9 | FAILS | 44.02±6.37 | 43.4±6.1 | |
| Mozabite | 25 | Yoruba (21) | Moroccan (22) | 23±1.9 | 21±1.3 | Long-range LD | -- | 21.6±1.8 | |
| Turkish | 17 | Mongola (10) | Iranian (13) | 28±3.2 | 24±1.5 | FAILS | 25.62±2.48 | 28.5±2.3 | |
| Brahui | 23 | Bantu Kenya (11) | Balochi (21) | 13±3.4 | 20±1.5 | Long-range LD | -- | 10.4±1.6* | Possibly multi-way admixture (*Pagani et al., 2017*) |
| Yemeni | 4 | Bantu Kenya (11) | Syrian (16) | 15±2.3 | 14±1.8 | FAILS | 6.29±2.89 | 12.7±1.6 | |
| Pima | 14 | Turkish (17) | Mayan (21) | 9±3.6 | 6±0.9 | SUCCEEDS | 6.29±0.89 | 7.8±1.1 | |
| Bantu South Africa | 8 | San Khomani (30) | Yoruba (21) | 26±2.5 | 25±2.3 | Long-range LD | -- | 27.9±2.2 | |
| Tu | 10 | Greek (20) | Han N-China (10) | 33±6.3 | 25±2.3 | FAILS | 28.83±2.8 | 31.3±1.96 | |
| West Sicilian | 10 | Yoruba (21) | East Sicilian (10) | 26±7.8 | 27±3.9 | FAILS | 42.72±16.34 | 37.4±16.4 | |
| Cambodian | 10 | Uyghur (10) | Han (34) | 17±4.7 | 20±2.7 | SUCCEEDS | 24.28±5.36 | 33.6±3.7* | |
| Georgian | 20 | Adygei (17) | Greek (20) | -- | 30±3.3 | FAILS | 3.15±1.22 | -- | |
| Romanian | 13 | Lithuanian (10) | East Sicilian (10) | -- | 31±2.6 | FAILS | -- | -- | |
| Bulgarian | 18 | Polish (16) | Cypriot (12) | -- | 28±3.5 | FAILS | 40.95±16.42 | 91.1±24.7* | Possibly multi-way admixture or different model of admixture (see Main text and *Haak et al., 2015*) |
| Hezhen | 8 | Tujia (10) | Mongola (10) | -- | 13±1.3 | FAILS | 2.92±1.4 | -- | |
| Oroqen | 9 | Yakut (25) | Mongola (10) | -- | 15±2 | Long-range LD | -- | -- | |
| Hungarian | 18 | Cypriot (12) | Polish (16) | 65±24 | 39±3.5 | FAILS | 54.83±25.27 | 61.8±19.1 | |
| Han N-China | 10 | Turkish (17) | Tujia (10) | 37±11.1 | 26±3.8 | FAILS | 48.17±10.36 | 44.3±5.1* | |
| Daur | 9 | Tujia (10) | Mongola (10) | -- | 21±1.7 | FAILS | -- | -- | |
| Greek | 20 | Polish (16) | Cypriot (12) | 69±18.5 | 36±3.7 | FAILS | 55.54±8.93 | 62.6±16.9 | |
| Melanesian | 10 | Papuan (16) | Cambodian (10) | 66±12.1 | 28±7.6 | FAILS | 64.91±5.42 | 68.6±7.1* | |
| Mandenka | 22 | Moroccan (22) | Yoruba (21) | 22±10.3 | 19±4.2 | FAILS | 17.25±6.05 | 85.8±19.0 *#Ω | Possibly multiple admixture events (*Price et al., 2009*) |
| Indian | 13 | Cambodian (10) | Sindhi (23) | 91±41.1 | 53±8.4 | FAILS | n/a | n/a | There are multiple "Indian" groups in the dataset making it unclear which target was used |
| North Italian | 12 | Cypriot (12) | French (28) | -- | 71±11.8 | FAILS | 12.44±4.32 | -- | |
| Polish | 16 | French (28) | Lithuanian (10) | -- | 31±5.1 | FAILS | -- | -- | |
| Tuscan | 8 | Cypriot (12) | French (28) | -- | 35±6.1 | FAILS | -- | -- | |
| San Namibia | 5 | Sandawe (28) | San Khomani (30) | -- | 48±8.9 | Long-range LD | -- | -- | |

*Appendix 1—table 5 Continued on next page*

*Appendix 1—table 5 Continued*

| Population | nk | Source1 | Source2 | ROLLOFF | Globetrotter | ALDER formal test | ALDER_2-ref dates | DATES | Comments |
|---|---|---|---|---|---|---|---|---|---|

Columns 1–5 include results from Table S12 from **Hellenthal et al., 2014**. We only show significant dates (|Z| > 2).

Following Hellenthal et al., we created a merged dataset of the Human Genome Diversity Panel, Henn et al. and Behar et al. containing 1642 individuals and 465543 SNPs. This dataset was used for ALDER and DATES analysis.

Standard errors in DATES were estimated using chromosome jackknife (see Materials and methods).

-- indicates results where the inferred results were not significant, either the method failed or the 95% CI included 0.

\* indicates DATES estimates that significantly differ from Globetrotter estimates (not within two SEs).

\# indicates DATES estimates that significantly differ from ROLLOFF results.

Ω indicates DATES estimates that significantly differ from ALDER results.

n/a indicates target population was unclear.

## Comparing *DATES* and modified ALDER for admixture times in Neolithic samples

We compared the performance of *DATES* with a modified version of ALDER used in *Lipson et al., 2017*, where they use a helper high coverage sample in order to infer the timing of admixture in three regions. Our results suggest that we obtain statistically consistent results for all overlapping samples (within two standard errors). Though an advantage of DATES is that it does require helper samples for the inference, as does modified ALDER. In some cases, higher coverage samples maybe available, though unless these samples have similar ancestry, they could bias the dates obtained.

**Appendix 1—table 6.** Admixture dates for Neolithic European groups using *DATES (Distribution of Ancestry Tracts of Evolutionary Signals)* and modified ALDER (*Lipson et al., 2017*).

| Region | Population | Modified ALDER* (Lipson et al., 2017) | DATES[†] |
|---|---|---|---|
| Germany | Blätterhöhle_MN | 18.5±4.6 | 14±3 |
| Germany | Germany_MN | 26.2±4.4 | 36±20 |
| Germany | LBK_EN | 14.9±2.4 | 18±3 |
| Hungary | LBK_EN | 17.8±2.0 | 22±2 |
| Hungary | Baden_CA | 27.6±3.8 | 49±11 |
| Hungary | Lasinja_CA | 29.3±5.2 | 32±6 |
| Hungary | LBKT_MN | 30.3±5.8 | 23±10 |
| Hungary | Protoboleraz_CA | 44.3±6.4 | 46±7 |
| Hungary | Starcevo_EN | 4.5±1.9 | 5±2 |
| Hungary | TDLN | 20.9±2.7 | 29±4 |
| Hungary | Tisza_LN | 18.2±6.6 | 27±9 |
| Spain | Iberia_CA | 49.6±5.2 | 55±6 |
| Spain | Iberia_EN | 19.4±2.3 | 20±3 |
| Spain | Iberia_MN | 49.9±7.7 | 52±8 |

*Modified ALDER – We report the individual level dates from Extended Data Table 4 based on average of individual level dates calculated using Anatolian farmers and western hunter-gatherer (WHG) as sources and high coverage Anatolian farmers as helper samples. For details, see *Lipson et al., 2017*

[†]DATES – We used pooled WHG groups and Anatolian farmers as references in DATES.

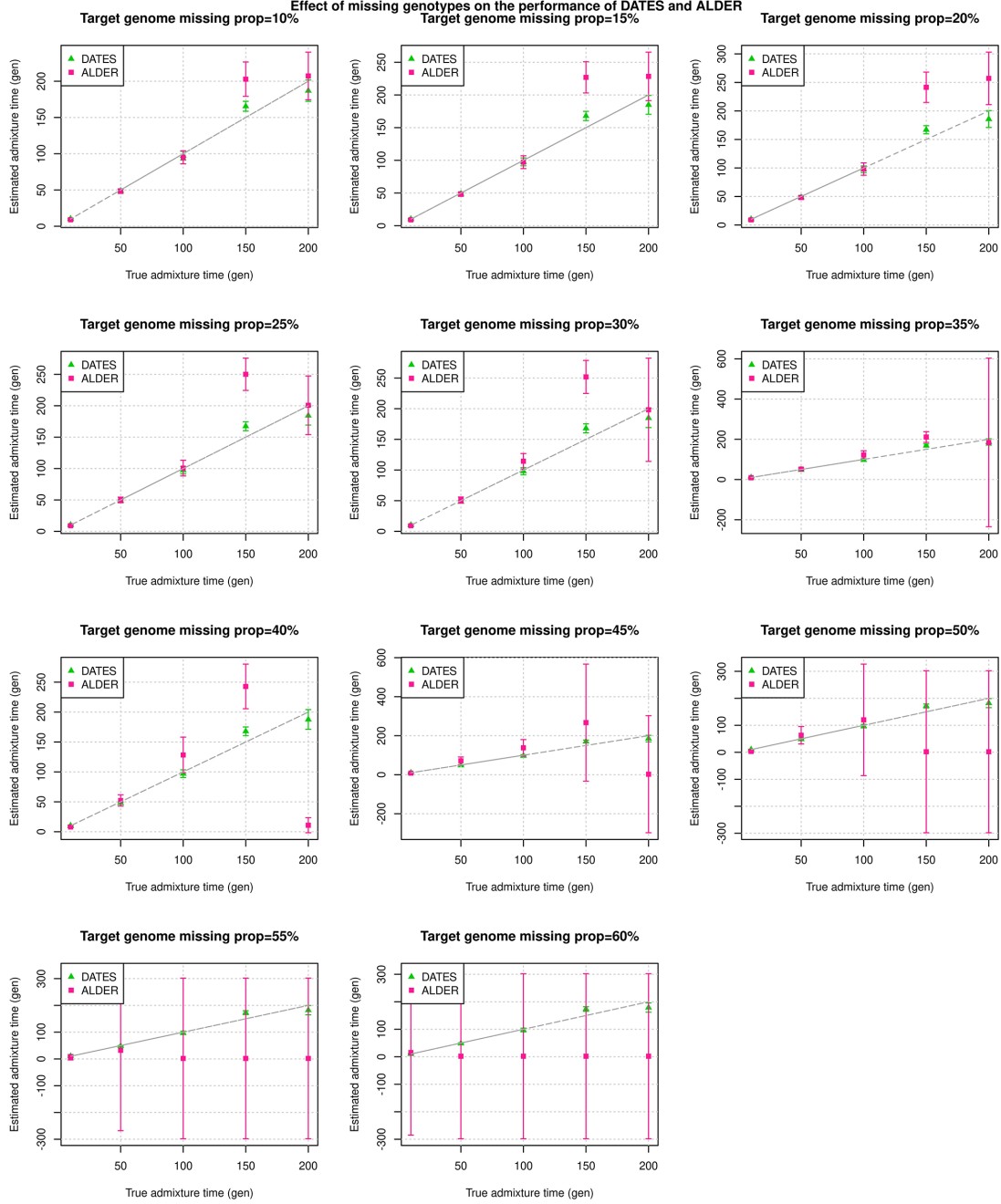

**Appendix 1—figure 5.** Effect of missing genotypes on the performance of *DATES (Distribution of Ancestry Tracts of Evolutionary Signals)* and ALDER: We simulated data for 10 admixed individuals with varying proportions of missing data (shown in each panel). The estimated admixture times (±1 SE) from *DATES* (green) and ALDER (pink) are shown on Y-axis and the true time of admixture is shown on X-axis. For a fair comparison with ALDER, the dates reported here for *DATES* are using exponential fit to $\lambda - 1$ generations (instead of the default of $\lambda$ generations).

## Appendix 2

### Simulations to test the performance of *DATES* under complex admixture scenarios involving founder events, multiple pulses of gene flow, or gradual admixture

*DATES* models admixed individuals as a mix of two source populations. However, in real data, the target could have ancestry from more than two source populations, or the gene flow can occur continuously over a period of time. To test the performance of *DATES* under these more complex scenarios, we performed additional simulations.

### Multiple pulses of gene flow

Using the simulation model described in Materials and methods, we generated a target population that has ancestry from three groups (*PopA*, *PopB*, *PopC*) that mixed at two distinct times ($t_1$ and $t_2$ generations ago) (**Appendix 2—figure 1**). Specifically, we generated data for three sets of admixed populations each with 10 individuals, where *PopA*, *PopB*, and *PopC* differ across runs. For each simulation, the older pulse of admixture occurred $t_2$ (=30, 60, 100) generations ago and *PopA* and *PopB* mixed with $\alpha_1/\alpha_2$ ancestry, respectively. This mixture was followed by additional gene flow from *PopC* that contributed $\alpha_3$ ancestry at $t_1$ (=10) generation ago. We used CEU, YRI, and CHB as *PopA*, *PopB*, and *PopC* and varied the order of the three ancestral populations to generate multiple sets of simulated individuals. We estimated admixture time for all three sets of simulated individuals using French, Tujia, and Yoruba from the HGDP dataset as reference populations.

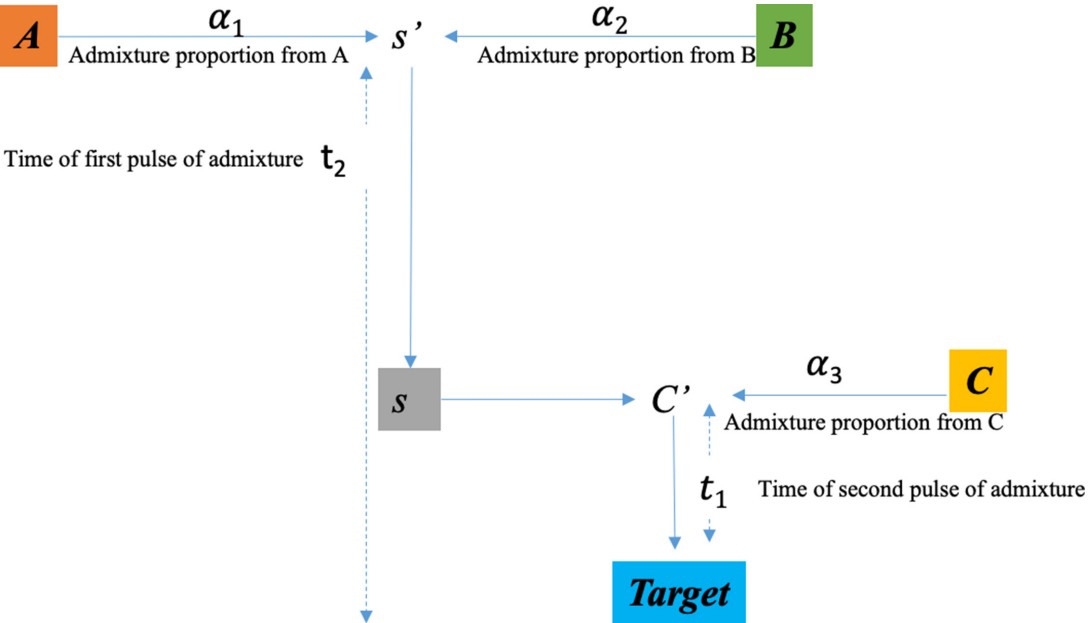

**Appendix 2—figure 1.** Model for multiple pulses of admixture. The admixed population (*Target*) derives ancestry from three populations, from the two gene flow events that occurred $t_2$ generations ago (older pulse) between *PopA* and *PopB* with $\alpha_1$ / $\alpha_2$ ancestries respectively resulting in an intermediate group *S*, which then mixes with *PopC* with $\alpha_3$ ancestry at $t_1$ generations ago (younger pulse).

### Equal proportions of ancestry from the three source populations

In this setup, we simulated admixed individuals with ancestry from *PopA*, *PopB*, and *PopC* with $\alpha_1 = 50\%$, $\alpha_2 = 50\%$, $\alpha_3 = 33\%$ thus the effective ancestry proportion in the admixed population would be 33% from each ancestral group. We varied the order of the three source populations and applied *DATES* to infer the timing of the mixture. Our results showed that when the references populations used for the inference correspond to the true admixing sources, in most cases, we reliably estimate the timing of both the pulses of admixture (**Appendix 2—figure 2**).

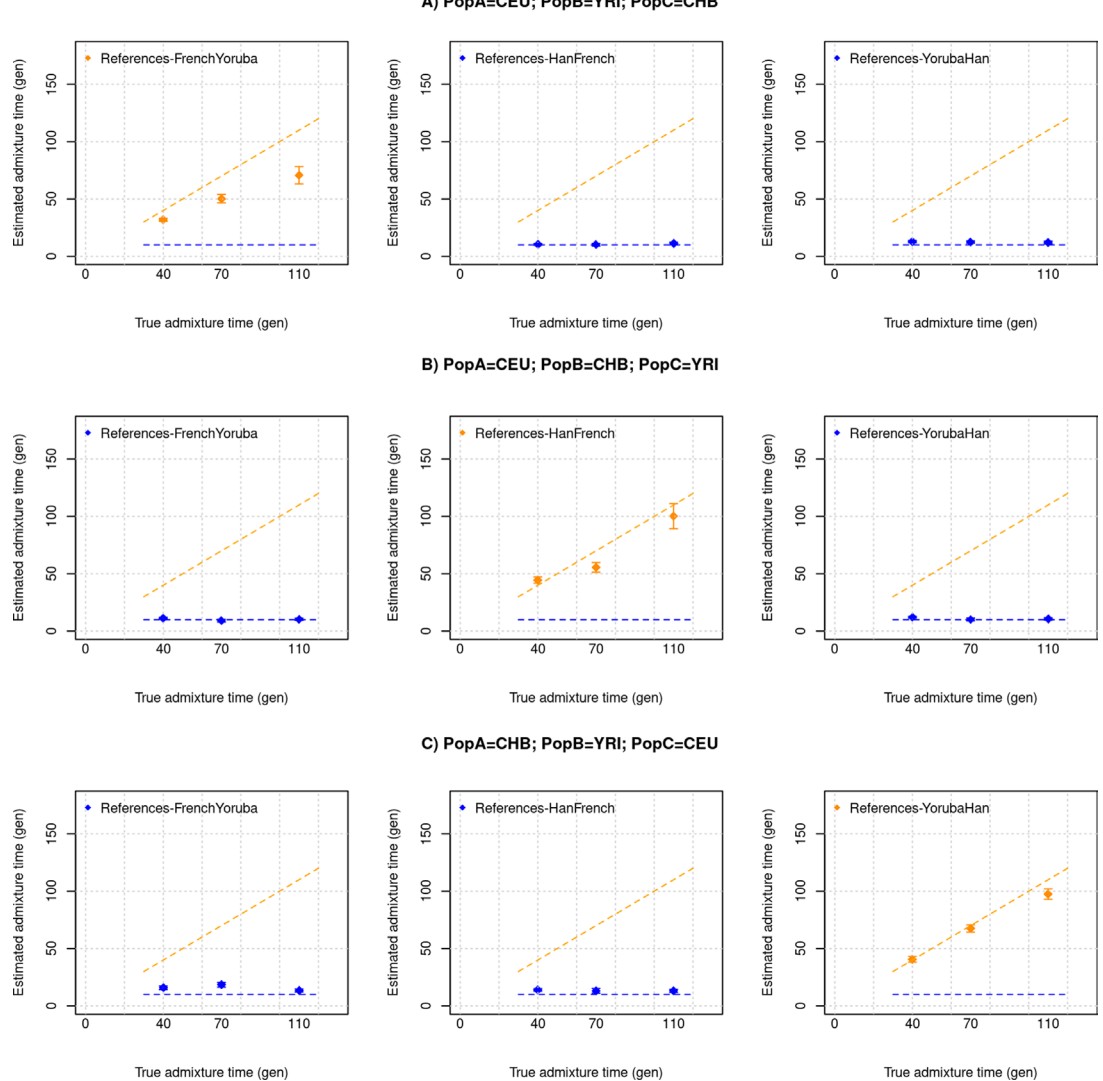

**Appendix 2—figure 2.** Multiple pulses of admixture with equal proportions of ancestry from sources. We generated a target population that has ancestry from three groups (*PopA*, *PopB*, *PopC*) with ancestry proportions of 33%, 33%, and 33% respectively that mixed at two distinct times ($t_1$=30,60,100 and $t_2$=10 generations ago). We used CEU, YRI, and CHB as *PopA*, *PopB*, and *PopC* and varied the order of the three ancestrals, and applied *DATES* with pairs of populations as the reference to infer the timing of the mixture. We show the expected dates ($t_1$ or $t_2$ depending on the references used), the orange dashed line corresponds to the older pulse and the blue dashed line corresponds to the younger pulse of admixture. The blue points correspond to *DATES* estimates using Pop*A* and Pop*C* or Pop*B* and Pop*C* as references. The orange points correspond to *DATES* estimates using Pop*A* and Pop*B* as references. Panel (**A**) shows the admixture scenario with PopA = CEU, PopB = YRI, and PopC = CHB. Panel (**B**) shows the admixture scenario with PopA = CEU, PopB = CHB, and PopC = YRI, and Panel (**C**) shows the admixture scenario with PopA = CHB, PopB = YRI, and PopC = CEU.

## Unequal proportions of ancestry from three ancestrals

In most real-world scenarios, the ancestry proportions of the admixing groups are unlikely to be exactly the same or similar. Thus, we generated data for groups with unequal proportions of ancestry from *PopA*, *PopB*, and *PopC* by setting $\alpha_1 = 20\%$, $\alpha_2 = 80\%$, and $\alpha_3 = 20\%$ or 80%. We varied the order of the three ancestral groups and applied *DATES* to infer the timing of the mixture. We observed that we recovered the timing of the recent pulse of admixture in most cases (***Appendix 2—figure 3***). In some cases, there was confounding in the timing of the recent event, when the % of ancestry from *PopC* was low (20%). In ***Appendix 2—table 1***, we explore how choosing ancestral populations that are more aligned with the model of admixture (that can be reliably inferred using other methods like *qpAdm*) can alleviate this bias (***Appendix 2—table 1***).

## Choice of references in a multi-pulse admixture

Using the admixture scenario described above, we examined how the choice of reference populations impacts the inferred dates of admixture. From *Appendix 2—figure 3*, we find that using PopC as one of the references gives more reliable dates. In most real-world scenarios, the ordering of gene flow events can be reliably inferred. For instance, for present-day Europeans, it's known that Steppe pastoralist gene flow occurred after the gene flow between Anatolian farmers and local European HGs. Thus, we fixed one reference group as *PopC* and explored how the choice of the second reference population impacts the recovery of the recent gene flow event. To ensure that our observed dates are not biased, we ran 10 replicates for each run and report the average of the estimated dates for the 10 runs below. Specifically, for the target populations with ancestries from *PopA*, *PopB*, and *PopC*, we ran *DATES* with the following reference populations:

1. Ref1: *PopC* and Ref2: *PopA.*
2. Ref1: *PopC* and Ref2: *PopB.*
3. Ref1: *PopC* and Ref2: admixed individuals with *PopA* and *PopB* ancestry ($30\%$ *PopA* / $70\%$ *PopB*) ancestry.
4. Ref1: *PopC* and Ref2: pooled samples from *PopA* and *PopB.*

We observed that in all four cases using *PopC* (the admixing source related to the most recent pulse) as one of the reference populations allows us to recover the timing of the recent pulse of admixture. When the other reference population is either the pooled set of samples of *PopA* and *PopB* (admixing sources of the older pulse) or admixed individuals with ancestry from *PopA* and *PopB*, we reliably infer the recent pulse of admixture in all cases (*Appendix 2—table 1*). In other cases, we observe some confounding in the inferred dates with the estimated dates falling intermediate to the two dates.

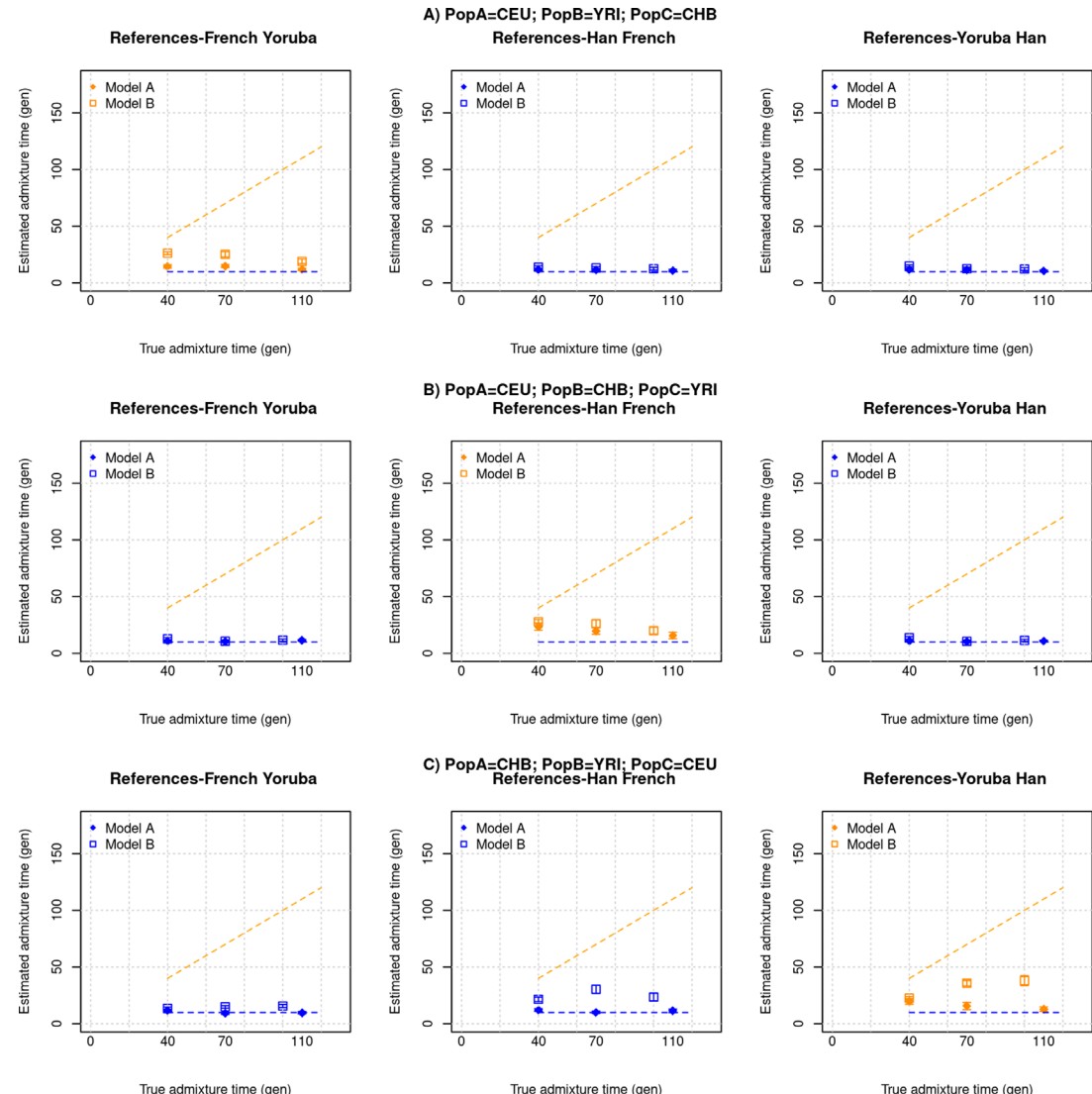

**Appendix 2—figure 3.** Two pulses of admixture with unequal proportions of ancestry from reference populations. We generated a target population that has ancestry from three groups with variation in ancestry from three sources. Model A: *PopA*, *PopB*, *PopC* with ancestry proportion of 4%, 16%, and 80% respectively that mixed at two distinct times ($t_1$=30,60,100 and $t_2$=10 generations ago). Model B: *PopA*, *PopB*, *PopC* with ancestry proportion of 16%, 64%, and 20% respectively that mixed at two distinct times ($t_1$=30,60,100 and $t_2$=10 generations ago). We used CEU, YRI, and CHB as *PopA*, *PopB*, and *PopC* and varied the order of the three ancestral populations, and applied *DATES* with pairs of populations as the reference to infer the timing of the mixture. Figures shows the true time of admixture on the X-axis and inferred time on Y-axis, the orange dashed line corresponds to the older pulse and the blue dashed line corresponds to the younger pulse of admixture. The blue points correspond to *DATES* estimates using Pop*A* and Pop*C* or Pop*B* and Pop*C* as references. The orange points correspond to *DATES* estimates using Pop*A* and Pop*B* as references. Panel (**A**) shows the admixture scenario with PopA = CEU, PopB = YRI, and PopC = CHB. Panel (**B**) shows the admixture scenario with PopA = CEU, PopB = CHB, and PopC = YRI. Panel (**C**) shows the admixture scenario with PopA = CHB, PopB = YRI, and PopC = CEU.

**Appendix 2—table 1.** Impact of reference populations in two-way admixed groups.

Model: We generated target populations with two pulses of gene flow where *PopA* and *PopB* mixed at time $t_1$ generations ago with ancestry proportion of $\alpha_1$ and $\alpha_2$, followed by gene flow from *PopC* at $t_2$ generations ago with ancestry proportion of $\alpha_3$

| Target | $t_2/t_1$ | Ref1 | Ref 2 | $\alpha_1$=20%, $\alpha_2$=80%, $\alpha_3$=80% | $\alpha_1$=50%, $\alpha_2$=50%, $\alpha_3$=50% | $\alpha_1$=20%, $\alpha_2$=80%, $\alpha_3$=20% | $\alpha_1$=20%, $\alpha_2$=80%, $\alpha_3$=10% |
|--------|-----------|------|-------|--------|--------|--------|--------|

*Appendix 2—table 1 Continued on next page*

*Appendix 2—table 1 Continued*

**Model: We generated target populations with two pulses of gene flow where *PopA* and *PopB* mixed at time $t_1$ generations ago with ancestry proportion of $\alpha_1$ and $\alpha_2$, followed by gene flow from *PopC* at $t_2$ generations ago with ancestry proportion of $\alpha_3$**

(A) Using reference populations *PopA* and *PopC*

| | | | | | | | |
|---|---|---|---|---|---|---|---|
| *PopA = CEU*<br>*PopB = YRI*<br>*PopC = CHB* | 100/10 | Han | French | 10.3 | 10.7 | 12.0 | 12.6 |
| *PopA = CEU*<br>*PopB = CHB*<br>*PopC = YRI* | 100/10 | Yoruba | French | 10.3 | 9.7 | 10.2 | 10.6 |
| *PopA = CHB*<br>*PopB = YRI*<br>*PopC = CEU* | 100/10 | French | Han | 10.8 | 11.6 | 20.7 | 44 |
| *PopA = CEU*<br>*PopB = YRI*<br>*PopC = CHB* | 60/10 | Han | French | 11.3 | 10.7 | 11.8 | 14.4 |
| *PopA = CEU*<br>*PopB = CHB*<br>*PopC = YRI* | 60/10 | Yoruba | French | 10.4 | 10.8 | 10.3 | 11.2 |
| *PopA = CHB*<br>*PopB = YRI*<br>*PopC = CEU* | 60/10 | French | Han | 11.2 | 12.6 | 19.9 | 40.7 |

(B) Using reference populations *PopB* and *PopC*

| | | | | | | | |
|---|---|---|---|---|---|---|---|
| *PopA = CEU*<br>*PopB = YRI*<br>*PopC = CHB* | 100/10 | Han | Yoruba | 10.2 | 11.5 | 11.3 | 11.9 |
| *PopA = CEU*<br>*PopB = CHB*<br>*PopC = YRI* | 100/10 | Yoruba | Han | 10.1 | 9.7 | 10.3 | 10.5 |
| *PopA = CHB*<br>*PopB = YRI*<br>*PopC = CEU* | 100/10 | French | Yoruba | 10.1 | 12.8 | 12.2 | 14 |
| *PopA = CEU*<br>*PopB = YRI*<br>*PopC = CHB* | 60/10 | Han | Yoruba | 11.0 | 12.6 | 12.2 | 14.6 |
| *PopA = CEU*<br>*PopB = CHB*<br>*PopC = YRI* | 60/10 | Yoruba | Han | 10.3 | 11.1 | 10.6 | 11.4 |
| *PopA = CHB*<br>*PopB = YRI*<br>*PopC = CEU* | 60/10 | French | Yoruba | 10.4 | 13.4 | 12.4 | 18 |

(C) Using reference populations *PopC* and 'admixed' individuals with ancestry from *PopA* and *PopB* (30% *PopA*/70% *PopB*) ancestry

| | | | | | | | |
|---|---|---|---|---|---|---|---|
| *PopA = CEU*<br>*PopB = YRI*<br>*PopC = CHB* | 100/10 | Han | Admixed (30% CEU/ 70% YRI) | 10.1 | 10.9 | 10.8 | 10.5 |
| *PopA = CEU*<br>*PopB = CHB*<br>*PopC = YRI* | 100/10 | Yoruba | Admixed (30% CEU/ 70% CHB) | 10.1 | 9.5 | 10.0 | 10 |
| *PopA = CHB*<br>*PopB = YRI*<br>*PopC = CEU* | 100/10 | French | Admixed (30% CHB/ 70% YRI) | 10.0 | 11.2 | 10.9 | 10.8 |
| *PopA = CEU*<br>*PopB = YRI*<br>*PopC = CHB* | 60/10 | Han | Admixed (30% CEU/ 70% YRI) | 11 | 11.3 | 10.9 | 11.8 |
| *PopA = CEU*<br>*PopB = CHB*<br>*PopC = YRI* | 60/10 | Yoruba | Admixed (30% CEU/ 70% CHB) | 10.3 | 10.8 | 10.3 | 10.5 |
| *PopA = CHB*<br>*PopB = YRI*<br>*PopC = CEU* | 60/10 | French | Admixed (30% CHB/ 70% YRI) | 10.2 | 11.3 | 10.6 | 12.1 |

(D) Using reference populations *PopC* and pooled individuals of *PopA* and *PopB* ancestry

| | | | | | | | |
|---|---|---|---|---|---|---|---|
| *PopA = CEU*<br>*PopB = YRI*<br>*PopC = CHB* | 100/10 | Han | French + Yoruba | 10.1 | 10.99 | 10.4 | 9.5 |

*Appendix 2—table 1 Continued on next page*

*Appendix 2—table 1 Continued*

**Model: We generated target populations with two pulses of gene flow where *PopA* and *PopB* mixed at time $t_1$ generations ago with ancestry proportion of $\alpha_1$ and $\alpha_2$, followed by gene flow from *PopC* at $t_2$ generations ago with ancestry proportion of $\alpha_3$**

| | | | | | | | |
|---|---|---|---|---|---|---|---|
| *PopA* = CEU<br>*PopB* = CHB<br>*PopC* = YRI | 100/10 | Yoruba | French + Han | 10.2 | 9.5 | 10.03 | 9.9 |
| *PopA* = CHB<br>*PopB* = YRI<br>*PopC* = CEU | 100/10 | French | Han + Yoruba | 9.9 | 10.4 | 10.6 | 10 |
| *PopA* = CEU<br>*PopB* = YRI<br>*PopC* = CHB | 60/10 | Han | French + Yoruba | 10.9 | 10.8 | 10.3 | 10.3 |
| *PopA* = CEU<br>*PopB* = CHB<br>*PopC* = YRI | 60/10 | Yoruba | French + Han | 10.3 | 10.7 | 10.1 | 10.5 |
| *PopA* = CHB<br>*PopB* = YRI<br>*PopC* = CEU | 60/10 | French | Han + Yoruba | 10.0 | 10.3 | 9.8 | 11 |

Note: the estimated dates are shown per scenario are averages of 10 simulations.

## Coalescent simulations

To evaluate the performance of *DATES* under demographic models involving gradual gene flow or founder events, we performed simulations using the coalescent simulator, *macs* (*Chen et al., 2009*). For all the simulations described below, we generated data for three populations (*PopA*, *PopB*, and *PopC*) for a region of 100 Mb with 22 replicates. The effective population size ($N_e$) of all three populations was assumed to be 12,500 with the mutation rate and recombination rate was assumed as $1.2 \times 10^{-8}$ and $1 \times 10^{-8}$ per base pair per generation, respectively (*Halldorsson et al., 2019*; *Jónsson et al., 2017*). The divergence time between population *A* and *B* was assumed to be 1800 generations, which translates to an estimated $F_{ST}$ (*PopA*, *PopB*) of 0.067. *PopC* was formed by admixture between *PopA* to *PopB* that occurred either continuously over a period of $\lambda$ generations or instantaneously at time *t*. We combined two haploid chromosomes at random to generate one diploid chromosome.

### Impact of continuous gene flow

To model continuous gene flow, we simulated a gradual mixture in *PopC* from *PopA*/*PopB* over a period of $\lambda$ $(= 5 - 60)$ generations, leading to 20%/80% *PopA*/*PopB* ancestry. Applying *DATES* to *PopC* with *Pop A* and *PopB* as the reference populations showed that the inferred time was intermediate between the start and end of the period of gene flow (*Appendix 2—table 2*). This is similar to the results of other admixture dating methods like Globetrotter, ALDER, and ROLLOFF (*Hellenthal et al., 2014*; *Loh et al., 2013*; *Moorjani et al., 2016*) and can be explained by the fact that continuous admixture leads to mixtures of exponential curves, and resolving the timing in such case can be challenging due to the well-known difficulty of fitting a sum of exponentials to data with even a small amount of noise (*Osborne and Smyth, 2016*).

*macs command line for continuous admixture with $\lambda$ =5 generations:*

```
macs 120 1e8 -t 6e-4 -r 5e-4 -I 3 50 20 50 -em 0.0002 2 1 2000 -em 0.0003 2 1
0 -ej 0.00032 2 3 -ej 0.036 1 3
```

**Appendix 2—table 2.** Impact of continuous gene flow.
The table shows true and inferred times of admixture in *PopC* using *PopA* and *PopB* used as the reference populations.

| The true period of continuous admixture, $\lambda$ generations | Inferred time of admixture (mean ±1 SE) is shown on Y-axis generations |
|---|---|
| 10–15 | 15±1 |
| 20–30 | 23±2 |

*Appendix 2—table 2 Continued on next page*

Appendix 2—table 2 Continued

| The true period of continuous admixture, $\lambda$ generations | Inferred time of admixture (mean ±1 SE) is shown on Y-axis generations |
|---|---|
| 40–60 | 53±3 |
| 40–100 | 64±4 |

## Impact of founder event/ bottleneck post-admixture

Many human populations have a history of founder events in their recent evolutionary past (*Tournebize et al., 2020*). A founder event generates long-range LD in the target population, which could in principle be spuriously inferred as admixture-related ancestry covariance, confounding the dates of admixture. To explore the effect of this scenario, we simulated *PopC* that has ancestry from *PopA* and *PopB* due to gene flow that occurred $T_A$ generations ago. Following admixture, *PopC* experienced a bottleneck that occurred $T_B$ (=10, 80, or 100) generations ago where the effective population size decreased from 12,500 to $N_B$ (=10, 100, 500, and 1000) for a duration of $D_B$ generations (=1, 5, or 10). After $T_B$, the population recovered to the original population size of 12,500 (*Appendix 2—figure 4A*). We applied *DATES* to *PopC* using *PopA* and *PopB* as reference populations and found that the estimated dates of admixture were accurate when the bottleneck was not extreme (*Appendix 2—figure 4B*). In the case of strong bottlenecks where $N_B$ is less than 100, we observed a downward bias in the estimated admixture time.

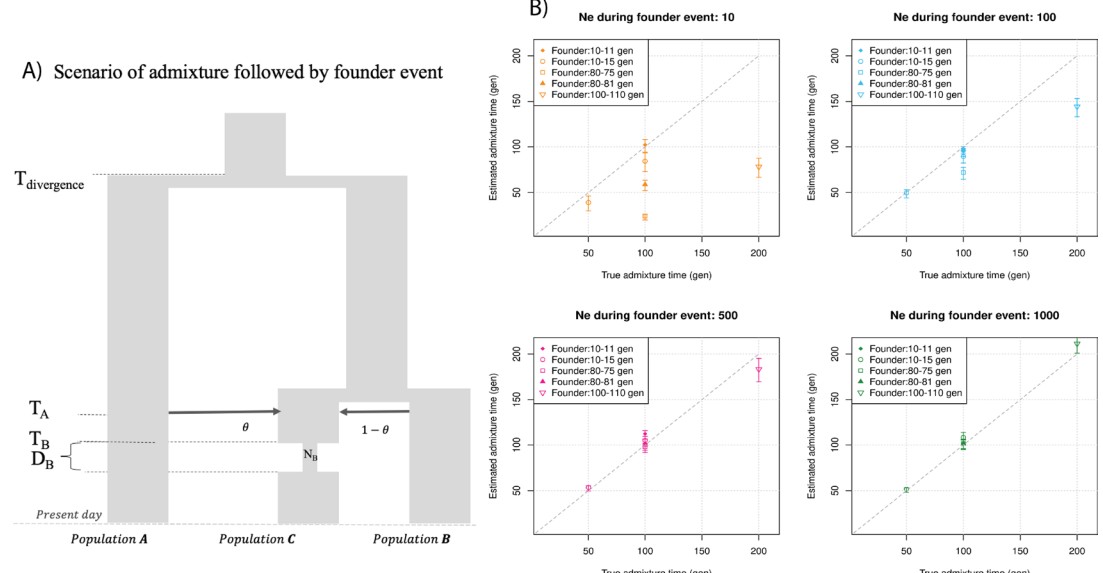

**Appendix 2—figure 4.** Impact of founder events on inferred dates of admixture. (A) Schematic for demographic scenario shows that *PopC* was formed through admixture between *PopA* and *PopB* at time $T_A$. Following admixture, *PopC* experienced a severe bottleneck that occurred $T_B$ generations ago where the effective population size decreased from 12,500 to $N_B$ for a duration of $D_B$ generations. After $T_B$, the population recovered to the original population size. (B) We simulated data for 10 individuals with admixture occurring at 50, 100, and 200 generations with bottleneck post-admixture for a period of $D_B$ = 1, 5, or 10 generations (shown in the legend) with the effective population size during bottleneck as $N_B$ = 10, 100, 500, or 1000 individuals (shown as four panels). The true admixture time is shown on X-axis, and the estimated time of admixture (±1 SE) is shown on Y-axis.

*macs command line:*

```
macs 120 1e8 -t 6e-4 -r 5e-4 -I 3 50 20 50 -em 0.002 2 1 10000 -em 0.00202 2
1 0 -en 0.0002 2 0.0002 -en 0.0003 2 1 -ej 0.00204 2 3 -ej 0.036 1 3
```

Another scenario we considered is when a population undergoes a severe bottleneck but does not recover (i.e., maintains a historically low population size to present). We simulated an admixed population that experienced a bottleneck post-admixture that occurred 100 generations ago.

The effective population size was then reduced from 12,500 to $N_B$ (=100–4000). This population maintained a small size until the present. Using *DATES* with the target as *PopC* and *PopA* and *PopB* as reference populations, we observed the inferred admixture times could be biased when $N_B < 1000$; there is no bias when the effective population size was larger (*Appendix 2—figure 5*, *Appendix 2—table 3*).

  *macs* command line:

```
macs 120 1e8 -t 6e-4 -r 5e-4 -I 3 50 20 50 -em 0.002 2 1 10000 -em 0.00202 2
1 0 -en 0 2 0.04 -en 0.00198 2 1 -ej 0.00204 2 3 -ej 0.036 1 3
```

**Appendix 2—table 3.** Admixture time estimates from *DATES (Distribution of Ancestry Tracts of Evolutionary Signals)* for populations with extreme bottlenecks with a historically low population size that does not recover until the present.

| Admixture | Ne before admixture | Ne post-admixture | Inferred time of admixture |
|---|---|---|---|
| 100 | 12,500 | 4000 | 96±5 |
| | | 3500 | 96±5 |
| | | 3000 | 88±4 |
| | | 2500 | 99±5 |
| | | 2000 | 92±6 |
| | | 1500 | 78±6 |
| | | 1000 | 86±6 |
| | | 500 | 54±9 |
| | | 100 | 42±10 |

## Scenario of drift post admixture

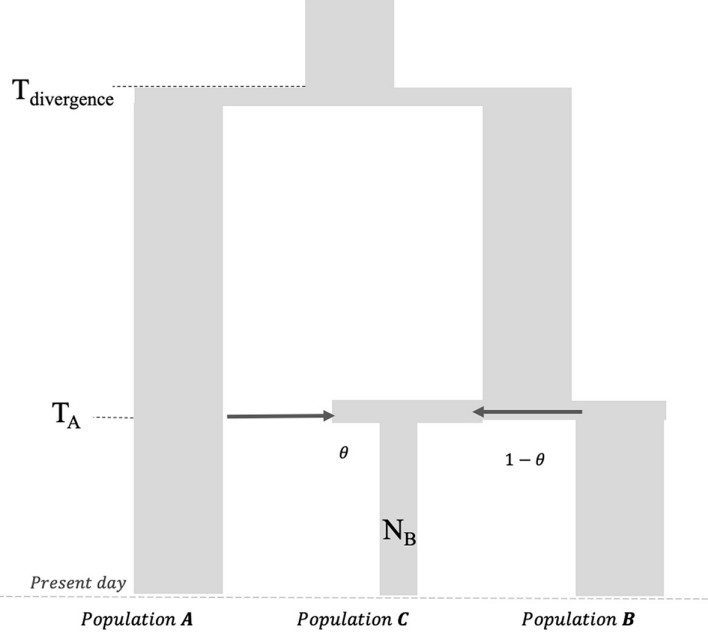

**Appendix 2—figure 5.** Impact of founder event with no recovery in admixed population. Schematic for the demographic history of the admixed group *PopC* that has ancestry from *PopA* and *PopB* followed by a severe bottleneck post-admixture without recovery to present (i.e., maintenance of historically low population size to present $N_B$). The $F_{ST}$ (*PopA, PopC*) is 0.202 and $F_{ST}$ (*PopB, PopC*) is 0.168.

## Simulations with no admixture in the target population

To investigate if *DATES* gives spurious results for admixture in the absence of gene flow from the reference populations, we generated data for populations without a history of recent admixture. We simulated individuals for three populations *PopA*, *PopB*, and *PopC*, where the divergence between *PopA* and *PopB* was 1800 generations and divergence between *PopC* and *PopB* was 1000 generations. *PopC* had a bottleneck that occurred $T_B$ (=100 or 10) generations ago where the population size reduced to $N_B$ (=100 or 10). Applying *DATES* with *PopC* as the target with *PopA* and *PopB* as reference populations, we observed no evidence of ancestry decay in *PopC* – the ancestry covariance curves were noisy and the 95% CI for the dates included 0 (***Appendix 2—figure 6***).

*macs command line:*

```
macs 120 1e8 -t 6e-4 -r 5e-4 -I 3 50 20 50 -en 0.0002 2 0.0002 -en 0.0003 2 1
-ej 0.02 2 3 -ej 0.036 1 3
```

Further, we simulated a target population with a severe bottleneck without recovery to present (without any admixture), where *PopC* had a bottleneck at 100 or 500 generations ago and maintained a small population size until present. The effective population size reduced from 12,500 to 1000 or 650 during this period (***Appendix 2—figure 7***). Using *DATES* on *PopC* as a target with *PopA* and *PopB* as reference populations, we observed that the ancestry covariance curves were noisy and the dates were not significant. This shows that *DATES* does not provide spurious evidence for admixture even for populations that have a complex history with strong founder events.

*macs command line:*

```
macs 120 1e8 -t 6e-4 -r 5e-4 -I 3 50 20 50 -en 0 2 0.013 -en 0.00202 2 1 -ej
0.00204 2 3 -ej 0.036 1 3
```

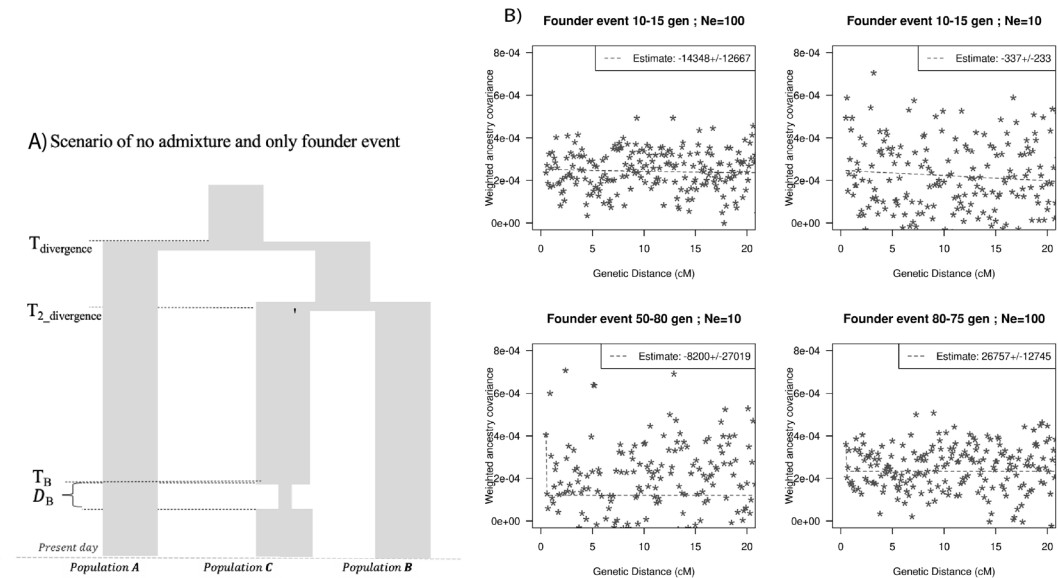

**Appendix 2—figure 6.** Impact of models with no admixture, severe founder event. (**A**) Demographic scenario with three populations. *PopA* and *PopB* diverged 1800 generations ago and *PopB* and *PopC* diverged 1000 generations ago. *PopC* had a bottleneck at $T_B$ generations ago with population size during bottleneck $N_B$. (**B**) Ancestry covariance curves for *PopC*. We simulated data for 25 individuals from *PopA* and *PopB* and 10 individuals from *PopC*. We applied *DATES (Distribution of Ancestry Tracts of Evolutionary Signals)* on *PopC* with *PopA* and *PopB* as sources and show the decay curves for different timing and effective population size of founder events. Note, none of the simulations show significant exponential fits and all dates include 0 in the estimated CI.

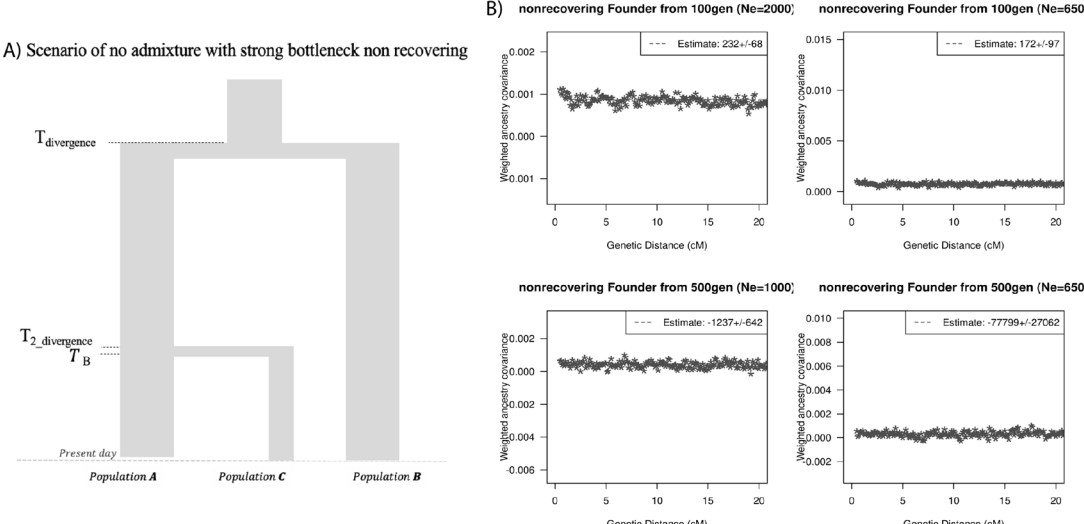

**Appendix 2—figure 7.** Effect of drift, no admixture: impact of models with no admixture, severe founder event (without recovery). (**A**) Demographic scenario with three populations. *PopA* and *PopB* diverged 1800 generations ago and *PopB* and *PopC* diverged 1000 generations ago. *PopC* had a bottleneck at $T_B$ generations ago with population size during bottleneck $N_B$. The population size $N_B$ is maintained to present. (**B**) Ancestry covariance curves for *PopC*. We simulated data for 25 individuals from *PopA* and *PopB* and 10 individuals from *PopC*. We applied *DATES (Distribution of Ancestry Tracts of Evolutionary Signals)* on *PopC* with *PopA* and *PopB* as sources and show the decay curves for different timing and effective population size of founder events.

