## [Editor Report]

This manuscript presents DATES, a robust method to infer the timing of admixture events using genetic data from present-day or ancient (with paleogenomics data) individuals. In the study, DATES is applied to >1000 ancient human genomes to characterize major admixture events during the European Holocene. This work will be of interest to scholars in the fields of population genetics, paleogenomics, archeology, biological anthropology, and history.

---

## [Decision Letter]

**Decision letter after peer review:**

Thank you for submitting your article "The spatiotemporal patterns of major human admixture events during the European Holocene" for consideration by *eLife*. Your article has been reviewed by 2 peer reviewers, and the evaluation has been overseen by George Perry as the Reviewing and Senior Editor. The reviewers have opted to remain anonymous.

Essential revisions: I am in consensus with the reviewers that new simulations better reflecting the properties of the data will be essential for the revision. All other comments/requests for clarification also seem reasonable to me; thus, please see and respond to all of the reviewer comments provided below.

*Reviewer #1 (Recommendations for the authors):*

– My main concern with this manuscript is that the main text lacks details about the differences between this version of DATES and the one introduced in Narasimhan et al. (2019). I would strongly suggest authors incorporate some information from Supplementary File 10 into the main text.

– Would it be possible to explain a bit how the simulations are done? In particular, it is unclear to me how the different admixture times are simulated. I see that Moorjani et al. 2011 is cited as a reference for that, but it would be great to have some details about it in the manuscript.

– On page 4, you assess the use of Khomani San instead of Yoruba as a reference population (FST=~0.1). That is using a population that is divergent from the true admixture source. In this case, both populations are present-day samples. In aDNA studies, sometimes we see present-day populations being used as surrogates for ancestral populations in admixture studies. If possible, could you please provide some insight about how much this could affect DATES estimates and whether your method is robust to that?

– Can you please clarify why using the admixed samples themselves as one of the reference populations works reliably (beginning of page 5)?

*Reviewer #2 (Recommendations for the authors):*

This paper would be substantially strengthened by the inclusion of simulation results which mirror the properties of the empirical data; especially with respect to Fst between reference populations, and average sample sizes, non-contemporaneous sampling, data missingness and pseudo-haploid genotypes in both the target and reference populations.

It would also be very useful for potential users of DATES for the authors to specify what they think are the lower bounds for the use of their method with respect to sample size and data quality.

There are some issues of consistency in the main text in the way the age estimates are presented, with many results lacking supporting information such as sample sizes and +/- standard errors

In the extended data figures, some of the population names are inconsistent with the labels present in the supplementary tables, making it difficult to follow exactly which samples were used. Please check that all the population names are consistent with the labels in Supplementary_File_5.

[Editors’ note: further revisions were suggested prior to acceptance, as described below.]

Thank you for resubmitting your work entitled "The spatiotemporal patterns of major human admixture events during the European Holocene" for further consideration by *eLife*. Your revised article has been evaluated by George Perry (Senior and Reviewing Editor).

I am satisfied with your major revision efforts, but there are a few points from the reviewers that you addressed in the response to reviewers document, but not in the manuscript itself, which was the intention.

Specifically, these three points:

1. On page 4, you assess the use of Khomani San instead of Yoruba as a reference population (FST=~0.1). That is using a population that is divergent from the true admixture source. In this case, both populations are present-day samples. In aDNA studies, sometimes we see present-day populations being used as surrogates for ancestral populations in admixture studies. If possible, could you please provide some insight about how much this could affect DATES estimates and whether your method is robust to that?

2. Can you please clarify why using the admixed samples themselves as one of the reference populations works reliably (beginning of page 5)?

3. It would also be very useful for potential users of DATES for the authors to specify what they think are the lower bounds for the use of their method with respect to sample size and data quality.

---

## [Author Response]

Reviewer #1 (Recommendations for the authors):– My main concern with this manuscript is that the main text lacks details about the differences between this version of DATES and the one introduced in Narasimhan et al. (2019). I would strongly suggest authors incorporate some information from Supplementary File 10 into the main text.

We agree with the reviewer. We have now added a section in the Methods which describes the differences in the model and implementation of the old and new versions of *DATES*.

– Would it be possible to explain a bit how the simulations are done? In particular, it is unclear to me how the different admixture times are simulated. I see that Moorjani et al. 2011 is cited as a reference for that, but it would be great to have some details about it in the manuscript.

We have now added detailed description of how the simulator works in the Methods.

– On page 4, you assess the use of Khomani San instead of Yoruba as a reference population (FST=~0.1). That is using a population that is divergent from the true admixture source. In this case, both populations are present-day samples. In aDNA studies, sometimes we see present-day populations being used as surrogates for ancestral populations in admixture studies. If possible, could you please provide some insight about how much this could affect DATES estimates and whether your method is robust to that?

In our analysis, we do not use present-day reference samples for dating events in ancient DNA samples. In *DATES*, reference samples are mainly used for inferring the allele frequencies in the ancestral populations of the target individuals. As long as the allele frequencies in the presentday samples are correlated to the ancestral allele frequencies, the use of present-day samples should not bias the inference. However, recent demographic events (e.g., strong founder events or admixture from another source, etc.) could have occurred in the history of the present-day samples that can lead to significant changes in allele frequencies, making present-day samples a poor surrogate of the allele frequencies in the ancestral populations.

– Can you please clarify why using the admixed samples themselves as one of the reference populations works reliably (beginning of page 5)?

Admixed individuals have mosaic ancestry from the relevant admixing source populations and in turn they have allele frequencies that are intermediate to the frequencies in the two source populations. Thus, they can provide a useful source of ancestral allele frequencies, especially in cases where genetic data from all ancestral mixing groups is not available. Moreover, Loh et al. 2013 showed that the use of admixed populations as one of the references does not bias the rate of decay of the weighted ancestry covariance (i.e., time of admixture), though the amplitude of the decay curve can be impacted (not used in our analysis). Admixed samples can thus be used for the inference (albeit with reduced power).

Reviewer #2 (Recommendations for the authors):This paper would be substantially strengthened by the inclusion of simulation results which mirror the properties of the empirical data; especially with respect to Fst between reference populations, and average sample sizes, non-contemporaneous sampling, data missingness and pseudo-haploid genotypes in both the target and reference populations.

We thank the reviewer for this suggestion. We note that the parameters of the simulations were chosen to match the estimates in empirical data. In empirical analysis, the *F_ST_* values between the admixing populations is fairly large. Specifically, the *F_ST_*(western hunter-gatherers, Anatolian farmers)=0.123, *F_ST_*(western hunter-gatherers, Steppe pastoralists)=0.109 and, *F_ST_*(Anatolian farmers, Steppe pastoralists)=0.055. We also have large sample sizes from all reference populations (*n*=16–60, average *n*=37) and moderate sizes for target populations (*n*=1–71, average *n*=8) (Supplementary file 1A).

To address the reviewers concern, we have now added a new simulation (Figure 1 – supplement 8 and 9) which includes combinations of features highlighted by the reviewer. We also note that two previous simulations also investigated combined effects of data limitations in ancient DNA (Figure 1—figure supplement 5 and Figure 1—figure supplement 7).

Figure 1 – supplement 8 and 9 : We simulated data for *n* admixed individuals with 20% European (CEU) and 80% African (YRI) ancestry with pseudo-haploid genotypes. The references used for inference are also pseudo-haploid genotyped. We further varied three key features of the data, missing genotype rate in reference populations (between 0-40%), missing genotype rate in target populations (between 20-60%) and divergence between true source populations and reference population used in *DATES*. We evaluated the performance of DATES for this setup for (supplement 8) target population size of *n* = 10 and (supplement 9) target population size of *n* = 1.

We have added the following to the main text:

“We also generated data for combinations of features including small sample sizes, pseudohaploid genotypes with large proportions of missing genotypes in both target and reference samples and use of highly divergent reference samples. We found *DATES* yielded reliable results even with large amounts of missing data (~40-60%) in the target or references individuals and use of highly divergent reference populations (Figure 1—figure supplement 8). This holds true even when using only a single target sample for dating but as expected with increasing missing data or larger divergence between the reference and true source populations, the inference becomes noisier (Figure 1—figure supplement 9).”

It would also be very useful for potential users of DATES for the authors to specify what they think are the lower bounds for the use of their method with respect to sample size and data quality.

While knowing the boundary conditions of an approach is useful for study design, we find it is hard to define lower bounds for inference, especially with the use of ancient DNA where the quality of data varies widely across individuals depending on sample preservation or sequencing coverage of the specimen. Moreover, the power of the inference depends on the demographic history, availability of data from reference populations related to the true source populations and age of the target samples (in particular, the proximity of the target sampling age to the time of admixture). Based on simulations, we find *DATES* gives reliable results using a single sample, with diploid or pseudo-haploid genotypes for least 50,000 genome-wide SNPs and with at least 5 samples from each reference population (with *F_ST_* ≤ 0.1). In empirical data, we find there is substantial robustness for dating recent admixture events, compared to older mixture events even with limited data. Thus, rather than suggesting lower bounds, we provide good of fit metrics (e.g., NRMSD, block jackknife, etc.) to evaluate the reliability of the inferred parameters.

There are some issues of consistency in the main text in the way the age estimates are presented, with many results lacking supporting information such as sample sizes and +/- standard errors

In the main text, we report average (or median where specified) dates of admixture, though details of all dates are provided in Figure 3 and Supplementary file 1B including sample size, average and standard errors in generations and years.

In the extended data figures, some of the population names are inconsistent with the labels present in the supplementary tables, making it difficult to follow exactly which samples were used. Please check that all the population names are consistent with the labels in Supplementary_File_5.

We thank the reviewer for noting these inconsistencies. We have fixed the sample/ population names in all tables and figures.

[Editors’ note: further revisions were suggested prior to acceptance, as described below.]

I am satisfied with your major revision efforts, but there are a few points from the reviewers that you addressed in the response to reviewers document, but not in the manuscript itself, which was the intention.Specifically, these three points:1. On page 4, you assess the use of Khomani San instead of Yoruba as a reference population (FST=~0.1). That is using a population that is divergent from the true admixture source. In this case, both populations are present-day samples. In aDNA studies, sometimes we see present-day populations being used as surrogates for ancestral populations in admixture studies. If possible, could you please provide some insight about how much this could affect DATES estimates and whether your method is robust to that?

We have added the following to the main text:

We found DATES is robust to the use of highly divergent surrogates as reference populations. For example, the use of Khomani San as the reference population instead of the true ancestral population of Yoruba (FST ~ 0.1) provides unbiased dates of admixture (Figure 1—figure supplement 4). In this regard, for ancient DNA where sometimes only sparse data is available, one can also use present-day samples as reference populations to increase the quality and sample size of the ancestral groups. In principle, as long as the allele frequencies in the reference samples are correlated to the ancestral allele frequencies, the inference of admixture dates should remain unbiased (Methods). In practice, however, recent demographic events (e.g., strong founder events or admixture from additional sources, etc.) in the history of the present-day samples could lead to significant deviation from the ancestral allele frequencies. Thus, the reference populations should be carefully chosen.

2. Can you please clarify why using the admixed samples themselves as one of the reference populations works reliably (beginning of page 5)?

We have added the following to the main text:

“Another idea is to use the admixed populations themselves as one of the reference populations as demonstrated by the single reference setup in ALDER (Loh et al., 2013; Pickrell et al. 2012). Admixed individuals have intermediate allele frequencies to the ancestral populations and thus weighted LD or ancestry covariance can be computed with only one reference population (albeit, with reduced power). Loh et al., 2013 showed that the use of admixed populations as one of the references does not bias the rate of decay of the weighted LD (i.e., time of admixture), though the amplitude of the decay curve (not used in DATES) can be biased under some scenarios. To verify DATES provides reliable results under this setup, we performed simulations and applied DATES with a single reference population and used the admixed population as the other reference. Like ALDER, our inferred dates of admixture were unbiased (Figure 1—figure supplement 6).”

3. It would also be very useful for potential users of DATES for the authors to specify what they think are the lower bounds for the use of their method with respect to sample size and data quality.

Lower bounds for statistical methods typically derive from theoretical limits of the method. Simulations can guide about the failure modes and serve as recommendations for parameter values that are likely to work, though these should not be considered as hard limits. Moreover, all our simulations are based on demographic and evolutionary parameters in humans, though the method is applicable (and being applied) to data from other species that differ in diversity, recombination map, data quality and demographic history compared to humans. Thus, rather than suggesting lower bounds, we have provided users with detailed investigation of the performance of DATES for varying sample size and data quality of reference and target individuals. Specifically, we provide recommendations for the following parameters in the main text under the “Overview of DATES: Model and simulations” section. Also where useful, we have added additional clarifications and caveats.

a) Dates of admixture – lines 137-139

b) Sample size of the target population – lines 140-141

c) Sample size of the reference populations – line 142

d) The genetic divergence of the source populations from the true ancestral population – lines 163-166

e) The quality of the data, including the proportion of missing genotypes in reference and target groups- lines 186-189.